# GRAPH IS A NATURAL REGULARIZATION: REVISITING VECTOR QUANTIZATION FOR GRAPH REPRESENTATION LEARNING

## ABSTRACT

Vector Quantization (VQ) has recently emerged as a promising approach for learning discrete representations of graph-structured data. However, a fundamental challenge, i.e., codebook collapse, remains underexplored in the graph domain, significantly limiting the expressiveness and generalization of graph tokens. In this paper, we present the first empirical study showing that codebook collapse consistently occurs when applying VQ to graph data, even with mitigation strategies proposed in vision or language domains. To understand why graph VQ is particularly vulnerable to collapse, we provide a theoretical analysis and identify two key factors: early assignment imbalances caused by redundancy in graph features and structural patterns, and self-reinforcing optimization loops in deterministic VQ. To address these issues, we propose RGVQ, a novel framework that integrates graph topology and feature similarity as explicit regularization signals to enhance codebook utilization and promote token diversity. RGVQ introduces soft assignments via Gumbel-Softmax reparameterization, ensuring that all codewords receive gradient updates. In addition, RGVQ incorporates a structure-aware contrastive regularization to penalize the token co-assignments among dissimilar node pairs. Extensive experiments demonstrate that RGVQ substantially improves codebook utilization and consistently boosts the performance of state-of-the-art graph VQ backbones across multiple downstream tasks, enabling more expressive and transferable graph token representations.

## 1 INTRODUCTION

In recent years, a discretization-based tokenization method, known as Vector Quantization (VQ), has attracted significant research attention for its effectiveness in generative modeling (Van Den Oord et al., 2017; Caron et al., 2018). VQ quantizes continuous latent representations into discrete clusters referred to as "codewords" in a learnable codebook (Zhang et al., 2023). These codewords are then trained to reconstruct the original data samples. By discretizing the latent space, VQ provides an effective prior for learning disentangled features, and has achieved remarkable success in various generative modeling tasks, including image synthesis (Ramesh et al., 2021; Chang et al., 2023; Li et al., 2024a), speech generation (Dhariwal et al., 2020; Zhang et al., 2024), and language models (Liu et al., 2025; Van Baalen et al., 2024).

Motivated by these successes, recent efforts have begun to explore the extension of VQ to graphs for scalable and versatile graph tokenization. First, discretizing graphs into VQ tokens enables compact graph compression, substantially reducing the memory and computation overhead during inference (Yang et al., 2023; Luo et al., 2024). Second, VQ provides a natural mechanism for abstracting structural patterns into a reusable token vocabulary, analogous to the language tokens used in Large Language Models (LLMs), and offers a promising pathway toward Graph Foundation Models (GFMs) (Wang et al., 2024). Third, VQ allows graphs to be serialized into token sequences, enabling sequence-based modeling with standard Transformer architectures that are widely adopted in NLP and vision, and eliminating the need for handcrafted inductive biases that are typically required in Graph Transformers (Wang et al., 2025).

Similar to VQ models in the vision and language domains, which are typically trained to reconstruct input samples (Navaneet et al., 2024; Deng et al., 2024), Graph VQ is also trained with reconstruction objectives, i.e., node feature and link reconstruction. Nevertheless, through our empirical study, we observe that **codebook collapse** consistently occurs during the training, even when applying mitigation strategies that are commonly used in other domains. This refers to the phenomenon where most inputs are mapped to only a few codewords, leaving the majority underutilized (Zhu et al., 2024; Zhang et al., 2023). As a result, only a limited number of tokens can be utilized during inference, leading to overly coarse representations and significant degradation in performance. However, prior work focuses primarily on the performance of downstream tasks, without addressing this critical problem in the first place (Wang et al., 2024; Zeng et al., 2025; Yang et al., 2023). This consistent underutilization naturally raises a central question: *What makes Graph VQ more prone to collapse?*

To answer this question, we investigate the phenomenon from both empirical and theoretical perspectives. Empirically, we find that the severity of collapse is correlated with typical graph properties such as structural and feature redundancy, indicating that inherent characteristics of graph data can aggravate the issue. Based on this observation, we theoretically identify two underlying factors: (1) early assignment imbalances caused by similar features and local structures; and (2) self-reinforcing training dynamics in VQ models, where frequently assigned codewords receive more updates and become increasingly dominant, while rarely selected ones remain inactive. This amplifies assignment imbalances and ultimately leads to collapse.

Based on these insights, we propose **Regularized Graph Vector Quantization (RGVQ)**, a novel framework that integrates graph topology and feature similarity as explicit regularization signals to enhance codebook utilization. First, to break the self-reinforcing loops, RGVQ adopts the Gumbel-Softmax reparameterization to relax hard assignments into differentiable probability distributions, enabling the gradients to flow not only to the most likely codewords but also to the less probable candidates. Second, RGVQ leverages graph topology and feature similarity to regularize token assignment distributions, explicitly penalizing token co-assignments induced by graph redundancy. This regularization encourages nodes with similar features and local structures to share token distributions, while discouraging similar assignment among unrelated nodes. Our contributions can be summarized as follows.

- To the best of our knowledge, this is the first study to systematically investigate the problem of codebook collapse in graph data, a fundamental but underexplored challenge in discrete graph token learning.

- We conduct an empirical study on codebook collapse in Graph VQ and provide a theoretical analysis of its root causes from both data and optimization perspectives.

- We propose RGVQ, a novel framework that effectively addresses early token co-assignment bias by structure-aware regularization and disrupts the self-reinforcing dynamics in VQ training via stochastic quantization.

- We perform comprehensive experiments on state-of-the-art (SOTA) Graph VQ backbones, demonstrating that our proposed method improves codebook utilization and downstream performance, and serves as a general solution for graph token learning.

## 2 RELATED WORK

**Vector Quantization**. Vector Quantization (VQ) maps continuous inputs to discrete tokens in a codebook and has been widely used in image, video, and audio generation (Chung et al., 2020; Fifty et al., 2024; Tang et al., 2022). This success has motivated efforts to extend VQ to graph data. For example, VQ-GNN (Ding et al., 2021) and VQGraph (Yang et al., 2023) apply VQ in supervised node classification tasks for embedding compression. However, their fully supervised training deviates from the original unsupervised training scheme of VQ (Chen & Lee, 2021; Yu et al., 2021). More recently, GFT pretrains VQ by reconstructing graph features to utilize the learned codebook as transferable vocabulary across tasks and domains (Wang et al., 2024). GQT employs residual VQ to tokenize graphs for vanilla transformers, alleviating manual architectural bias in graph transformers (Wang et al., 2025). While both methods demonstrate promising applications of Graph VQ, they overlook the issue of codebook collapse, which undermines the generalization of learned tokens. HQA-GAE introduces a hierarchical VQ with annealing for codeword selection,

improving performance on graph tasks (Zeng et al., 2025). Nevertheless, it does not effectively resolve the non-differentiability of VQ and lacks a formal analysis of codebook collapse.

**Collapse Mitigation**. One of the most fundamental limitations of VQ is codebook collapse, wherein only a small fraction of codewords are used (Zhang et al., 2024; Lu et al., 2023). Various mitigation strategies have been explored. For example, Exponential Moving Average (EMA) is proposed to stabilize codebook updates (Polyak & Juditsky, 1992; Wu & Yu, 2019). Pretraining the encoder (Zhao et al., 2024) is proposed to mitigate embedding drift during training VQ. In addition, codebook reset (Zeghidour et al., 2021; Williams et al., 2020) periodically reinitializes inactive codewords with encoder embeddings. Affine parameters (Huh et al., 2023; Zhang et al., 1997) introduce a learnable transformation before quantization to align encoder outputs with the codebook space. Recently, SimVQ (Zhu et al., 2024) reparameterizes the code vectors through a linear transformation layer based on a learnable latent basis. Although these mitigation strategies have been evaluated in image and speech domains, their performance on graph data remains underexplored.

## 3 PRELIMINARY

**Graph Neural Network**. Graph Neural Networks (GNNs) learn the node representations by recursively aggregating features from neighbors, also known as message-passing (Sun et al., 2022; Zhang et al., 2019). Formally, the representation of node $v$ at the $l$-th layer is:

$$h_v^{(l)} = \text{AGG}(\{h_u^{(l-1)}, u \in \mathcal{N}(v) \cup v\}, \phi^{(l)}), \tag{1}$$

where $h_v^{(0)} = x_v$ is the initial node feature, $\mathcal{N}(v)$ is the neighbor set of node $v$, and $\phi^{(l)}$ is the parameters of the $l$-th layer of the GNN. The aggregation function $\text{AGG}(\cdot)$ combines the embedding of node $v$ and its neighbors, which is typically implemented as sum, mean, or max pooling.

**Deterministic VQ**. VQ maps continuous vectors into a finite set of discrete embeddings in the codebook (Van Den Oord et al., 2017). Given a codebook $\mathbf{C} = \{e_i\}_{i=1}^K$ with each discrete codeword $e_i \in \mathbb{R}^d$, a continuous input $h_i \in \mathbb{R}^d$ is quantized as $z_i$ with the nearest codebook vector $e_k$ by:

$$k = \arg\min_j \|h_i - e_j\|_2^2 = \arg\min_j \|h_i - \delta_j \mathbf{C}\|_2^2, \tag{2}$$

where $\delta_j \in \{0, 1\}^{1 \times K}$ is the one-hot indicator vector with only the $j$-th element being 1. To enable gradient propagation through the non-differentiable vector $\delta_j$, the Straight-Through Estimator (STE) is applied (Bengio et al., 2013). During the backward process, the gradient of the quantized embedding $z_i = \delta_j \mathbf{C}$ is copied to $h_i$, which is denoted as

$$z_i = \text{sg}(\delta_j \mathbf{C} - h_i) + h_i, \quad \Rightarrow \frac{\partial z_i}{\partial h_i} = 1, \tag{3}$$

where $\text{sg}[\cdot]$ denotes the stop-gradient operator, ensuring the gradient for $\delta_j \mathbf{C}$ is disgarded during the backward process. Finally, the learning objective is to reconstruct the input samples, with a codebook loss that pulls the quantized representations $\mathbf{Z} = \{z_1, z_2, \ldots, z_N\}$ toward the encoder outputs $\mathbf{H} = \{h_1, h_2, \ldots, h_N\}$, and a commitment loss that pulls the encoder outputs toward the quantized representations:

$$\mathcal{L}_{\text{VQ}} = \underbrace{\mathcal{L}_{\text{recon}}}_{\text{reconstruction loss}} + \underbrace{\|\text{sg}[\mathbf{H}] - \mathbf{Z}\|^2}_{\text{codebook loss}} + \beta \underbrace{\|\mathbf{H} - \text{sg}[\mathbf{Z}]\|^2}_{\text{commitment loss}}. \tag{4}$$

For Graph VQ, the reconstruction task typically involves reconstructing the graph properties, i.e., node features and links (Wang et al., 2024; 2025; Yang et al., 2023):

$$\mathcal{L}_{\text{recon}} = \underbrace{\frac{1}{N} \left\|\mathbf{X} - \hat{\mathbf{X}}\right\|_2^2}_{\text{feature reconstruction}} + \underbrace{\left\|\mathbf{A} - \hat{\mathbf{A}}\right\|_2^2}_{\text{link reconstruction}}, \tag{5}$$

where $\mathbf{A} \in \mathbb{R}^{N \times N}$ denotes the adjacency matrix, $N$ is the total number of nodes, and $\mathbf{X} \in \mathbb{R}^{N \times D}$ is the node feature matrix. The reconstructed feature matrix $\hat{\mathbf{X}} = g_{\theta_1}(\mathbf{Z})$ and the reconstructed adjacency matrix $\hat{\mathbf{A}} = g_{\theta_2}(\mathbf{Z})$ are generated by the decoders for the feature reconstruction and link reconstruction tasks, respectively (Wang et al., 2024; Zeng et al., 2025).

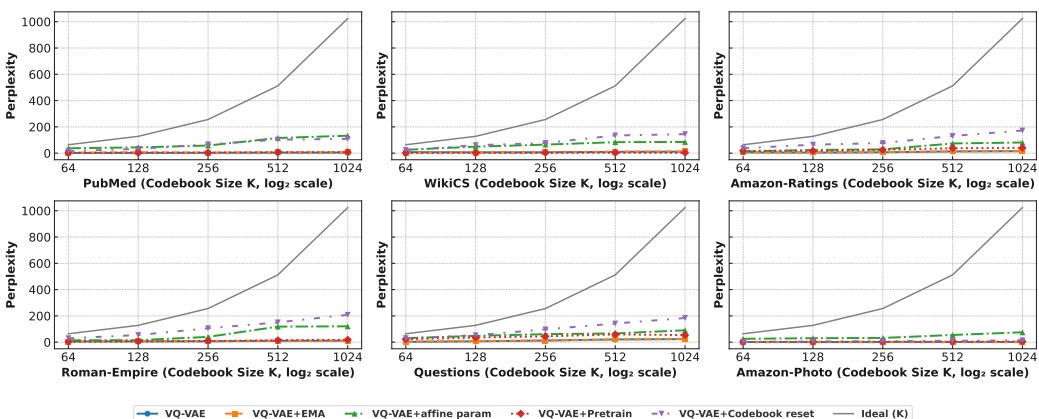

Figure 1: Codebook perplexites on graph datasets. The black lines indicate the optimal perplexities, i.e., codebook size K.

**Metric for Codebook Utilization**. The extent of codebook collapse is measured by the codebook perplexity (Takida et al., 2022; Yan et al., 2024; Zheng & Vedaldi, 2023), which is defined as:

$$P = \exp\left( -\sum_{k=1}^{K} p_k \log p_k \right),  \quad (6)$$

where $p_k$ denotes the probability of selecting the $k$-th codeword. A low perplexity indicates that only a few codewords dominate the assignments, reflecting a high degree of collapse. In contrast, a high perplexity suggests better utilization of the codebook capacity.

## 4 MOTIVATION

Recent studies demonstrate the potential of Graph VQ, while most focus on downstream performance, leaving codebook utilization largely underexplored. In this section, we conduct an empirical study of the codebook utilization and observe that codebook collapse occurs consistently during training, even when applying mitigation strategies from the language and vision domains. Moreover, we find that the severity of collapse correlates with the redundancy of graph data. Building on these observations, we provide a theoretical analysis and identify two key factors contributing to collapse: token co-assignments induces by similar features and local structures, and the self-reinforcing training dynamics of VQ models. These insights motivate the development of our method.

### 4.1 EMPIRICAL STUDY

We begin by investigating the codebook perplexity of Graph VQ on different graph datasets. Following the settings of prior work (Ding et al., 2021; Wang et al., 2024; Yang et al., 2023), we apply vanilla Graph VQ and its variants augmented with widely adopted collapse mitigation methods, including EMA, codebook reset, pretrained encoder, and affine parameters. By default, orthogonal normalization (Yu et al., 2021) and cosine similarities (Wang et al., 2024)

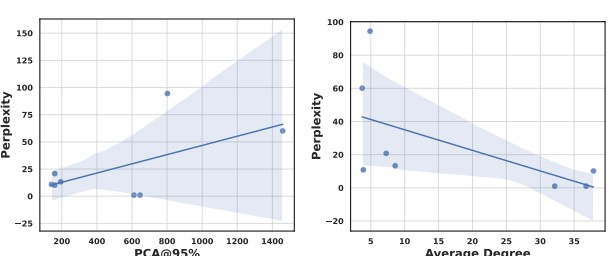

(a) PCA@95% vs. Perplexity    (b) Avg. Degree vs. Perplexity

Figure 2: Correlation between graph properties and codebook perplexity.

are incorporated in all variants. The implementation details and additional datasets can be found in Appendix B and C, respectively. From Figure 1, we make the following observations: **(Ob. 1)** Codebook collapse is a systematic and severe issue in Graph VQ. Across all datasets, the perplexity of VQ

remains far below the codebook capacity, and fails to grow proportionally with the increasing codebook size. **(Ob. 2)** General mitigation strategies adopted from other domains only achieve marginal improvements and fail to fundamentally address codebook collapse in graphs. These findings reveal that codebook collapse is not merely incidental, but a systematic issue in Graph VQ. We hypothesize that the unique properties of graph data, i.e., inherent feature redundancy and non-i.i.d. nature, may contribute to this phenomenon. To investigate this, we analyze two graph-level statistics that serve as proxies for these properties on investigated datasets. We consider: (1) PCA@95%, which quantifies feature redundancy by measuring the number of principal components needed to preserve 95% of node feature variance (Dong et al., 2022; Hou et al., 2023); and (2) average node degree, which reflects local connectivity density. Higher degrees imply stronger dependencies between neighboring nodes, violating the i.i.d. assumption and serving as a simple proxy for the non-i.i.d. nature of graph data (Yang et al., 2023; Wu et al., 2019). From Figure 2, we observe a positive correlation between PCA@95% and codebook perplexity, and a negative correlation between average degree and perplexity. Lower PCA@95% suggests higher feature redundancy, and higher average degree implies stronger local connectivity and non-i.i.d. characteristics, both associated with a greater tendency toward collapse.

## 4.2 THEORETICAL ANALYSIS

**Assignment Imbalances**. To better understand the mechanism behind the above observation, we provide a theoretical analysis that links graph characteristics to token assignment behavior. A key manifestation of codebook collapse is that nodes are increasingly mapped to the same codebook entry, regardless of their underlying differences. Thus, we begin by analyzing how the probability of two nodes being assigned to the same token is influenced by the similarity in terms of their features and local structures.

**Theorem 1** (Token Co-assignment Probability with GNN Encoder). *Given a L-layer GNN encoder $\phi$ with parameters $\mathbf{W} = (\mathbf{W_1}, \mathbf{W_2})$, for two nodes $v_1$ and $v_2$ sampled uniformly at random from the graph, the probability that they are quantized to the same codeword $p := \mathbb{P}[z_{v_1} = z_{v_2}]$ satisfies the following lower bound:*

$$p \geq \alpha \left( 1 - \frac{2\mathcal{B}_x}{\delta_c} \Big[ C_1 + \sum_{\ell=1}^{L} C_2^{\ell} \mathbb{E}[D_\ell] \Big] \right), \tag{7}$$

*where $\alpha \in (0, 1]$ is the local quantization consistency constant, $\delta_c$ is the minimum codeword radius of the codebook, and $C_1 = C_\sigma \mathcal{B}_{\mathbf{W}_1}$ and $C_2 = C_\sigma C_\rho C_g \mathcal{B}_{\mathbf{W}_2}$ are constants related to GNN parameters. Here $C_\sigma, C_\rho, C_g$, are Lipschitz term of GNNs. $\mathcal{B}_x$ is the bounded norm of node feature $\mathbf{X}$, and $\mathcal{B}_{\mathbf{W}_1}, \mathcal{B}_{\mathbf{W}_2}$ denote bounded norm of $\mathbf{W_1}, \mathbf{W_2}$, respectively. $D_\ell = d_\ell \times d_{\ell-1} \times ... \times d_1$, and $d_\ell$ indicates the number of children of the l-layer computation trees.*

*Proof.* The proof can be found in Appendix A. □

*Remark 1.* Theorem 1 provides a theoretical perspective on observations in the empirical study, it identifies a feature-related term captured by the feature norm bound $\mathcal{B}_x$, and a structure-related term captured by the computation-tree expansion $\mathbb{E}[D_\ell]$ as two fundamental factors that can induce collapse in graphs. This bound is conservative and should not be interpreted as a predictor of the collapse trend.

**Self-Reinforcing Dynamics of Graph VQ**. Given that these data-induced imbalances exist, we further analyze how the optimization dynamics of Graph VQ progressively amplify them, ultimately driving the codebook collapse. In Graph VQ, the codebook $\mathbf{C}$ is updated only through the vocabulary loss (Zhu et al., 2024), i.e., the second term in Equation 4. The update is denoted as:

$$\mathbf{C}^{(t+1)} = \mathbf{C}^{(t)} - \eta \, \mathbb{E}_{h_i} \left[ \delta_k^\top \delta_k \, \mathbf{C}^{(t)} \right] + \eta \, \mathbb{E}_{h_i} \left[ \delta_k^\top \, h_i \right], \tag{8}$$

where $h_i$ is the embedding of node $v_i$, $\eta$ is the learning rate, and $\delta_k^\top \delta_k$ is the Kronecker delta matrix, defined as:

$$(\delta_k^\top \delta_k)_{ij} = \begin{cases} 1 & \text{if } i = j = k, \\ 0 & \text{otherwise.} \end{cases} \tag{9}$$

Figure 3: Overall framework of RGVQ. Note: red nodes represent the positive set, while green nodes denote negative samples.

This condition indicates that when the expectation $\mathbb{E}_{h_i}[\delta_k^\top \delta_k] = I$, i.e., every token is selected with equal probability $\frac{1}{K}$, each codebook entry is updated during training. However, according to Theorem 1, token assignments in graphs exhibit early imbalances due to the intrinsic feature and structural redundancies. As a result, frequently selected codewords are more frequently updated and continuously pulled towards the distribution of the output of the GNN, i.e., $\mathbf{h}$. On the other hand, the encoder outputs are simultaneously optimized towards the selected codewords via the commitment loss in Equation 4. This hard assignment and bidirectional attraction form a self-reinforcing "cocoon effect," which not only locks the encoder into preferring codewords, but also suppresses any possibility of unused codewords being reactivated.

## 5 METHODOLOGY

Our analysis identifies that both the redundancy inherent in graph data and the self-reinforcing training dynamics of VQ as key factors contributing to codebook collapse. To address these challenges, we propose RGVQ, a regularized Graph VQ framework designed to mitigate collapse. The overall framework is illustrated in Figure 3. RGVQ first replaces hard assignments with differentiable assignment distributions using Gumbel-Softmax reparameterization, enabling gradients to flow to all codewords proportionally to their assignment probabilities. Building on this, RGVQ leverages graph topology and feature similarity to regularize token assignment distributions, explicitly penalizing co-assignment induced by graph redundancy.

**Gumbel-Softmax Reparameterization**. In deterministic VQ, the training dynamics of hard assignments prevent gradient backpropagation to unselected codewords, ultimately leaving them inactive and underutilized. To address this issue, we adopt Gumbel-softmax reparameterization (Roy et al., 2018; Sønderby et al., 2017), which replaces hard nearest-neighbor assignment with a differentiable soft selection. Formally, the assignment distribution of quantizing $h_i$ to entries in codebook $C$ is defined as:

$$p_i(\mathbf{C} \mid h_i) = \text{Softmax}(\pi_i), \quad \text{where } \pi_i = -\|h_i - \mathbf{C}\|^2. \tag{10}$$

Instead of using a non-differentiable argmax over the distribution, we apply the Gumbel-Softmax trick to estimate a differentiable approximation of this hard assignment. Specifically, the assignment distribution is perturbed with Gumbel noise and passed through a temperature-controlled softmax:

$$\tilde{p}_i(\mathbf{C} \mid h_i) = \text{Softmax}_\tau \left( \log p_i(\mathbf{C} \mid h_i) + g_i \right), \tag{11}$$

where $g_i \sim \text{Gumbel}(0,1)$ is the sampled noise from Gumbel (0,1) and $\tau$ is the temperature. Given this estimated distribution, the quantized embedding is computed as a weighted average over all codebook entries:

$$\tilde{z}_i = \sum_{j=1}^{K} \tilde{p}_i(e_j|h_i) \cdot e_j, \quad \text{where } e_j \in \mathbf{C}. \tag{12}$$

This soft assignment allows all codebook entries to receive gradient updates proportionally to their participation in the quantized representation, thus mitigating the pre-mentioned self-reinforced loop of deterministic VQ. During inference, the model reverts to deterministic hard assignment by selecting the codeword with the maximum logit $j = \arg\max_j p_i(\mathbf{C}|h_i)$.

---

**Algorithm 1** Training Procedure for RGVQ

---

**Input:** Encoder $f_\phi$, Decoder $g_\theta$, Codebook $\mathbf{C} = \{e_k\}_{k=1}^K$,
        Temperature $\tau$, commitment weight $\beta$.
**Output:** Model parameters $\theta, \phi$ and Codebook $\mathbf{C}$.

---

1: Initialize codebook $\mathbf{C}$ with K-means method.
2: Compute the positive set $\mathcal{N}_P$ and negative set $\mathcal{N}_P$ for every node.
3: **repeat**
4:     Sample minibatch $x \sim p_{\text{data}}$;
5:     $h = f_\theta(x)$;
6:     Distances to every codeword: $\pi = -\|h - \mathbf{C}\|^2$;
7:     Assignment distribution: $p(\mathbf{C}|h) = \text{Softmax}(\pi)$;
8:     Gumbel-softmax reparameterization: $\tilde{p}(\mathbf{C}|h) = \text{Softmax}_\tau(\log p(\mathbf{C}|h) + g)$;
9:     Soft quantization: $\tilde{z} = \sum_{j=1}^K \tilde{p}(\mathbf{e}_j|h)\,\mathbf{e}_j$;
10:    Minimize loss:

$$\mathcal{L} = \mathcal{L}_{\text{recon}} + \mathcal{L}_{\text{reg}} + \|\text{sg}[h] - \tilde{z}\|^2 + \beta\|h - \text{sg}[\tilde{z}]\|^2;$$

11: **until** converged

---

**Structure-Aware Regularization**. To mitigate token co-assignments induced by the redundancy of graph data, we incorporate feature and structural similarities to regularize the token assignment distribution, encouraging the model to avoid overuse of specific codebook entries. Our key insight is that collapse arises when nodes are spuriously mapped to the same tokens due to overly similar computation-tree embeddings caused by structural or feature redundancy. Therefore, we explicitly distinguish between similar and dissimilar node pairs based on both feature and structural similarity: similar nodes can exhibit more consistent assignment distributions, while dissimilar nodes should be discouraged from co-assignments. Formally, for a given anchor node $v$, we define:

- **Positive set** $\mathcal{N}_P$: consists of the $n$ sampled nodes that are either structurally or semantically similar to $v$. Specifically, $n$ positive nodes are sampled from the union of the following two candidate sets: (1) nodes directly connected to $v$; or (2) the top $K$ feature-similar nodes to $v$. Formally, the positive set is denoted as:

$$\mathcal{N}_P = \left\{ u \,\middle|\, (a_{uv} = 1) \,\vee\, \left(u \in \arg\text{topk}_{u' \in \mathcal{V}} \text{sim}(x_u, x_v)\right); \forall v \in \mathcal{V} \right\}, \quad (13)$$

  where $a_{uv} \in A$ is the adjacency matrix, $\text{sim}(\cdot,\cdot)$ is the similarity function, $x_u$ is the feature vector of node $u$. We apply the cosine similarity as the similarity function.

- **Negative set** $\mathcal{N}_N$: consists of the $n$ sampled nodes that are neither structurally connected nor semantically similar to $v$. Formally, the negative set is defined as:

$$\mathcal{N}_N = \left\{ u \,\middle|\, (a_{uv} = 0) \,\wedge\, \left(u \notin \arg\text{topk}_{u' \in \mathcal{V}} \text{sim}(x_u, x_v)\right) \,\wedge\, (u \neq v), \,\forall v \in \mathcal{V} \right\}. \quad (14)$$

We encourage nodes in the positive set to have similar assignment distributions, while penalizing nodes in the negative set for having overlapping token distributions. Formally, given two nodes $v_i$ and $v_j$, we use $\tilde{p}_i$ and $\tilde{p}_j$ to represent their token assignment distributions $\tilde{p}_i(\mathbf{C} \mid h_i)$ and $\tilde{p}_j(\mathbf{C} \mid h_j)$ respectively. Then the distributions are regularized by an InfoNCE loss (You et al., 2021; Wu et al., 2021), which is defined as:

$$\mathcal{L}_i = -\log \frac{\sum_{j \in \mathcal{N}_P} \exp(\text{sim}(\tilde{p}_i, \tilde{p}_j))}{\sum_{j \in \{\mathcal{N}_P \cup \mathcal{N}_N\}} \exp(\text{sim}(\tilde{p}_i, \tilde{p}_j))}. \quad (15)$$

We sum $\mathcal{L}_i$ all nodes to obtain the final regulation loss, i.e., $\mathcal{L}_{\text{reg}} = \sum_N \mathcal{L}_i$. This regularization term is then added to the reconstruction loss in Equation 4, forming the ultimate loss:

$$\mathcal{L}_{\text{VQ}} = \mathcal{L}_{\text{recon}} + \|\text{sg}[\mathbf{H}] - \tilde{\mathbf{Z}}\|^2 + \beta\|\mathbf{H} - \text{sg}[\tilde{\mathbf{Z}}]\|^2 + \mathcal{L}_{\text{reg}}. \quad (16)$$

Table 1: Mean codebook utilization in perplexity on homophilous and heterophilous graphs with codebook size $K = 512$. **Bold** highlights the best performance.

| | Cora | PubMed | Citeseer | Photo | Computer | WikiCS | Ratings | Roman | Questions |
|---|---|---|---|---|---|---|---|---|---|
| Graph VQ | 94.47±8.65 | 4.14±1.03 | 60.09±5.59 | 1.00±0.00 | 1.00±0.00 | 10.18±2.13 | 13.29±2.89 | 10.84±3.48 | 20.78±3.65 |
| EMA | 91.68±9.17 | 5.12±1.46 | 55.15±6.73 | 1.00±0.00 | 1.00±0.00 | 11.27±3.36 | 9.12±2.33 | 6.20±3.24 | 14.15±3.51 |
| AP | 75.32±6.28 | 126.55±12.64 | 9.03±2.16 | 54.95±5.43 | 59.33±8.55 | 83.55±9.86 | 73.82±8.14 | 118.46±16.21 | 66.57±8.27 |
| Reset | 65.79±8.56 | 102.78±15.78 | 85.19±4.41 | 10.73±1.98 | 17.18±2.37 | 134.44±7.35 | 130.83±8.88 | 150.51±11.15 | 141.98±10.11 |
| PT | 60.57±10.25 | 6.17±1.12 | 138.98±10.54 | 3.78±1.37 | 2.94±1.27 | 3.10±1.31 | 37.65±5.76 | 14.49±2.52 | 58.99±8.34 |
| SimVQ | 40.09±6.53 | 23.96±2.56 | 38.11±6.67 | 37.29±4.85 | 40.47±6.54 | 45.90±7.35 | 16.08±4.11 | 42.22±8.34 | 21.71±5.27 |
| HQA-GAE | 130.06±5.52 | 164.77±14.15 | 93.67±11.32 | 166.32±10.98 | 114.08±10.15 | 98.73±7.82 | 92.17±8.66 | 89.05±8.23 | 72.86±7.79 |
| RGVQ | **211.69**±5.27 | **319.09**±10.40 | **188.17**±11.23 | **446.02**±15.82 | **413.10**±10.78 | **228.82**±5.96 | **200.93**±7.89 | **374.51**±11.13 | **250.79**±8.63 |

Table 2: Cross-domain and cross-task performance in the pre-training and fine-tuning setting. Metrics are reported in terms of ROC-AUC for Graph Classification and Accuracy for all other tasks. **Bold** highlight the best performance.

| Method | Node Classification | | | Link Classification | | Graph Classification | | |
|---|---|---|---|---|---|---|---|---|
| | Cora | PubMed | WikiCS | WN18RR | FB15K237 | HIV | PCBA | *Avg.* |
| GCN | 75.65±1.37 | 75.61±2.10 | 75.28±1.34 | 73.79±0.39 | 82.22±0.28 | 64.84±4.78 | 71.32±0.49 | 74.10 |
| GAT | 76.24±1.62 | 74.86±1.87 | 76.28±0.78 | 80.16±0.27 | 88.93±0.15 | 65.54±6.93 | 70.12±0.89 | 76.01 |
| GIN | 73.59±2.10 | 69.51±6.87 | 49.77±4.72 | 74.02±0.55 | 83.21±0.53 | 66.86±3.48 | 72.69±0.22 | 69.95 |
| DGI | 72.10±0.34 | 73.13±0.64 | 75.32±0.95 | 75.75±0.59 | 81.34±0.15 | 59.62±1.21 | 63.31±0.89 | 71.51 |
| BGRL | 71.20±0.30 | 75.29±1.33 | 76.53±0.69 | 75.44±0.30 | 80.66±0.29 | 63.95±1.06 | 67.09±1.00 | 72.88 |
| GraphMAE | 73.10±0.40 | 74.32±0.33 | 72.61±0.39 | 78.99±0.48 | 85.30±0.16 | 61.04±0.55 | 63.30±0.78 | 72.66 |
| GIANT | 75.13±0.49 | 72.31±0.53 | 76.56±0.88 | 84.36±0.30 | 87.45±0.54 | 65.44±1.39 | 61.49±0.99 | 74.68 |
| GFT | 78.35±1.07 | 73.39±1.68 | 79.13±0.32 | 90.87±0.25 | 89.89±0.27 | 72.16±1.69 | 72.74±1.23 | 79.50 |
| GFT + EMA | 79.44±0.89 | 74.01±1.57 | 78.94±0.41 | 90.58±0.43 | 89.75±0.19 | 72.39±1.52 | 73.04±1.01 | 79.73 |
| GFT + AP | 79.69±1.07 | 75.05±0.86 | 79.73±0.35 | 89.56±0.18 | 89.05±0.18 | 71.86±1.53 | 71.48±0.99 | 79.48 |
| GFT + Reset | 80.07±0.91 | 75.51±0.69 | 79.85±0.33 | 91.18±0.43 | 88.09±0.23 | 72.79±1.65 | 71.95±0.85 | 79.92 |
| GFT + PT | 78.57±0.86 | 74.12±1.05 | 72.75±1.72 | 88.63±0.15 | 88.45±0.17 | 71.01±1.74 | 73.73±1.12 | 78.18 |
| GFT + SimVQ | 77.61±0.73 | 76.41±1.28 | 76.57±0.68 | 82.72±0.53 | 82.03±0.35 | 66.57±1.35 | 69.90±0.91 | 75.97 |
| GFT + RGVQ | **80.85**±0.73 | **77.46**±0.94 | **80.10**±0.52 | **91.32**±0.26 | **90.45**±0.31 | **74.10**±1.49 | **75.68**±0.99 | **81.42** |

## 6 EXPERIMENTS

To evaluate the generality and effectiveness of RGVQ, we conduct extensive experiments based on three key functions of Graph VQ: (1) codebook utilization, (2) transferability, and (3) serialization. First, we evaluate the codebook utilization of RGVQ on both homophilous and heterophilous graphs, aiming to verify its ability to mitigate codebook collapse. Second, we investigate the transferability. We integrate RGVQ into GFT (Wang et al., 2024), a graph foundation model that utilizes the learned codebook as pretrained graph tokens, and evaluate the performance on cross-task and cross-domain graphs. Third, we assess the serialization capability of RGVQ by examining its compatibility with sequence-based models. We use GQT (Wang et al., 2025), a transformer taking VQ tokens as input sequences, and evaluate the performance on node classification. Detailed dataset statistics, baselines, implementation details, and the complexity analysis are provided in Appendix B and D, respectively.

**Codebook Utilization**. We evaluate codebook utilization by comparing vanilla Graph VQ and its variants: EMA (Łańcucki et al., 2020), affine parameters (AP) (Huh et al., 2023), codebook reset (Reset) (Zeghidour et al., 2021), and pretrained encoders (PT) (Zhao et al., 2024), as well as existing SOTA VQ models: SimVQ (Zhu et al., 2024) and HQA-GAE (Zeng et al., 2025). All methods use orthogonal normalization, cosine similarity, and K-Means initialization (Im & Chan, 2023), and are trained on the feature and link reconstruction tasks. We fix the codebook size at 512 and report perplexity in Table 1. Across all datasets, RGVQ outperforms all baselines by a clear margin and remains robust, showing that Gumbel-Softmax reparameterization combined with structure-aware regularization leads to more balanced codebook use, prevents collapse, and enables more expressive graph representations. By contrast, vanilla Graph VQ and its variants suffer from severe codebook collapse, with perplexity values as low as 1.00 in several datasets. More advanced mitigation strategies like SimVQ and HQA-GAE offer only small improvements, indicating their limited ability to fundamentally address the underlying causes of collapse in Graph VQ.

**Transferability**. To evaluate the effectiveness of RGVQ in learning transferrable graph tokens, we integrate it into a graph foundation model, i.e., GFT, and compare it with vanilla Graph VQ

Table 3: Mean performance on node classification tasks. Metrics are reported in terms of ROC-AUC for Questions, and Accuracy for all other datasets. **Bold** indicates the best performance.

| | Cora | PubMed | Citeseer | Photo | Computer | WikiCS | Ratings | Roman | Questions |
|---|---|---|---|---|---|---|---|---|---|
| GCN | 75.65±1.37 | 78.80±0.60 | 71.60±0.40 | 92.70±0.20 | 89.65±0.52 | 77.47±0.85 | 48.70±0.63 | 73.69±0.74 | 76.09±1.27 |
| GAT | 76.24±1.62 | 79.00±0.40 | 72.10±1.10 | 93.87±0.11 | 90.78±0.13 | 76.91±0.82 | 52.70±0.62 | 88.75±0.41 | 76.79±0.71 |
| GraphGPS | 82.84±1.03 | 79.94±0.26 | 72.73±1.23 | 95.06±0.13 | 91.19±0.54 | 78.66±0.49 | 53.10±0.42 | 82.00±0.61 | 71.73±1.47 |
| SGFormer | 84.50±0.80 | 80.30±0.60 | 72.60±0.20 | 95.10±0.47 | 91.99±0.76 | 73.46±0.56 | 48.01±0.49 | 79.10±0.32 | 72.15±1.31 |
| Exphomer | 82.77±1.38 | 79.46±0.35 | 71.63±1.19 | 95.35±0.22 | 91.47±0.17 | 78.54±0.49 | 53.51±0.46 | 89.03±0.37 | - |
| NodeFormer | 83.20±0.90 | 79.90±1.00 | 72.50±1.10 | 93.46±0.35 | 86.98±0.62 | 74.73±0.94 | 43.86±0.35 | 64.49±0.73 | 74.27±1.46 |
| GQT | 86.44±1.58 | 81.60±1.35 | 73.14±1.26 | 94.46±0.68 | 92.13±0.23 | 80.03±0.19 | 54.04±0.12 | 89.85±0.73 | 76.52±1.52 |
| GQT + EMA | 86.23±1.19 | 81.41±1.24 | 73.08±1.58 | 94.01±0.57 | 91.95±0.18 | 79.98±0.23 | 54.10±0.08 | 89.91±0.51 | 75.94±1.16 |
| GQT + AP | 85.89±0.94 | 83.31±0.97 | 72.56±1.38 | 96.15±1.21 | 94.46±0.36 | 82.03±0.59 | 54.54±0.24 | 90.46±0.52 | 76.96±1.17 |
| GQT + Reset | 86.15±1.07 | 83.50±1.01 | 71.59±1.37 | 95.15±0.55 | 94.79±0.48 | 82.84±0.23 | 54.41±0.17 | 90.50±0.42 | 78.13±0.98 |
| GQT + PT | 85.71±1.44 | 80.92±1.15 | 79.53±1.23 | 94.74±0.76 | 92.35±0.35 | 75.65±0.78 | 54.50±0.14 | 89.76±0.68 | 76.74±1.34 |
| GQT + SimVQ | 86.02±1.64 | 82.56±1.02 | 72.58±1.14 | 95.21±0.77 | 94.23±0.21 | 81.78±0.32 | 53.98±0.15 | 90.15±0.66 | 76.35±1.21 |
| GQT + RGVQ | **88.34±1.32** | **86.54±1.41** | **81.25±1.01** | **97.66±1.05** | **95.67±0.36** | **83.58±0.66** | **55.16±0.19** | 90.98±0.66 | **78.26±1.07** |

Table 4: Drop-one ablation study.

| Variant | Gumbel-Softmax | Structure samples | Feature samples | Cora Perp. | Cora Acc. | PubMed Perp. | PubMed Acc. | WikiCS Perp. | WikiCS Acc. |
|---|---|---|---|---|---|---|---|---|---|
| Variant-1 | | | | 94.47 | 78.35 | 4.14 | 73.39 | 10.18 | 79.13 |
| Variant-2 | ✓ | ✓ | | 172.32 | 79.87 | 215.35 | 76.32 | 153.35 | 79.84 |
| Variant-3 | ✓ | | ✓ | 135.45 | 79.12 | 208.16 | 76.29 | 179.49 | 79.79 |
| RGVQ | ✓ | ✓ | ✓ | **211.69** | **80.85** | **319.09** | **77.46** | **228.82** | **80.10** |

and its variants with different mitigation strategies. Moreover, we include supervised GNNs, i.e., GCN (Zhang et al., 2022), GAT (Veličković et al., 2017), and GIN (Xu et al., 2018), and graph self-supervised methods, i.e., DGI (Veličković et al., 2018), BGRL (Thakoor et al., 2021), Graph-MAE (Hou et al., 2022), and GIANT (Chien et al., 2021). The supervised GNNs are trained directly on each target dataset, while the self-supervised methods and all GFT variants are pretrained on the full set of datasets and then fine-tuned per target. Table 2 reports the model performance across cross-domain and cross-task datasets. RGVQ consistently achieves the highest average performance across all tasks and datasets, surpassing both supervised and self-supervised graph models including GFT. These consistent gains indicate that alleviating codebook collapse is essential for improving the expressiveness and transferability of graph tokens. By encouraging more balanced and diverse codebook utilization, RGVQ learns discrete representations that generalize better across tasks and domains, positioning it as a general and robust solution for GFMs.

**Serialization**. To further evaluate the effectiveness of RGVQ in serialization, we integrate it into the Graph Quantized Transformer (GQT), where discrete tokens serve as the input sequence to a vanilla Transformer backbone. We follow the original sequence reconstruction method (Wang et al., 2025) and compare the performance of RGVQ-enhanced GQT against GQT with different anti-collapse methods, supervised GNNs, and graph transformers, including GraphGPS (Rampášek et al., 2022), SGFormer (Wu et al., 2023), Exphomer (Shirzad et al., 2023), and NodeFormer(Wu et al., 2022). Table 3 summarizes the node classification results across various benchmarks. Compared to GQT with conventional anti-collapse methods, incorporating RGVQ consistently improves classification accuracy on most datasets. These results highlight the importance of mitigating codebook collapse when learning discrete tokens for Transformer-based architectures.

## 6.1 PERFORMANCE EVALUATION

## 6.2 ABLATION STUDIES

We further conduct ablation experiments on the contributions of the Gumbel–Softmax reparameterization and the structure-aware contrastive regularization. We first compare RGVQ with three variants, each lacking a component in our design. Beyond the drop-one ablation, we also vary the temperature and the number of contrastive samples to assess the effect of each proposed module. Finally, we include the influence of codebook size to assess the robustness of our proposed model. We also include the additional ablation studies for each proposed module in Appendix C.

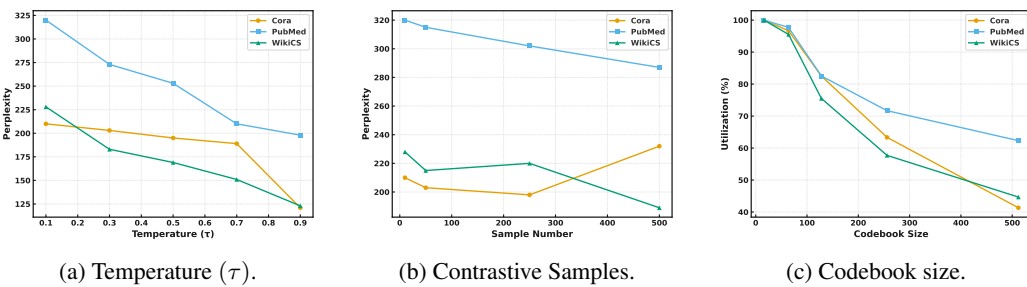

(a) Temperature ($\tau$).      (b) Contrastive Samples.      (c) Codebook size.

Figure 4: Ablation study with varying parameters.

**Drop-one Ablation**. As the structure-aware regularization relies on the soft assignment probabilities produced by the Gumbel–Softmax reparameterization; without reparameterization, one-hot assignments provide no gradient to inactive codewords, making the regularization ineffective. Conversely, using soft assignments without regularization does not constrain the assignment distribution and therefore makes RGVQ behave similarly to vanilla VQ. Therefore, we only assess three variants: removing all proposed components (variant-1), reparameterization with only structurally similar samples (variant-2), and reparameterization with only feature-similar samples (variant-3). As shown in Table 4, excluding either the structural-similar or feature-similar set reduces quantization diversity and consequently harms downstream accuracy. Removing Gumbel-softmax causes RGVQ to degenerate to vanilla VQ and leads to severe collapse. These results indicate that Gumbel-Softmax reparameterization and structure-aware regularization are mutually dependent in preventing codebook collapse, and both topological and feature information are essential for enhancing quantization diversity.

**Influence of the Temperature**. We investigate how the Gumbel–Softmax temperature $\tau$ affects perplexity. As shown in Figure 4(a), lower temperatures, which produce distributions closer to one-hot, consistently improve codebook utilization. This indicates that, unlike some prior work (Zeng et al., 2025) that rely on temperature annealing, a relatively low and fixed temperature is sufficient to regularize the codebook distributions and address the non-differentiability of deterministic VQ.

**Influence of Contrastive Samples**. We examine how the number of contrastive samples in structure-aware regularization affects perplexity. As shown in Figure 4(b), even a small number of contrastive samples (e.g., $n = 10$ or $50$) achieves relatively high perplexity, indicating effective codebook utilization. Increasing $n$ does not consistently lead to better utilization and may even degrade in some datasets, potentially due to added training noise.

**Influence of the Codebook Size**. We evaluate how codebook size affects the codebook utilization of RGVQ, which is defined as the ratio of utilized codebook entries to the total size of the codebook. The results are shown in Figure 4(c). Across all datasets, RGVQ consistently maintains high utilization as the codebook size increases, showing strong robustness to the choice of the codebook size. Notably, even with a large codebook size of 512, the model utilizes over 50% of the codebook capacity. This suggests that RGVQ can flexibly adapt to different representational granularities, and its token assignment remains stable even under large vocabulary settings.

## 7 CONCLUSION

In this paper, we investigate the codebook collapse problem in Graph VQ. Through empirical studies, we show that codebook collapse is not an incidental phenomenon, but a systematic issue of Graph VQ models. We provide a theoretical analysis of the underlying causes and propose RGVQ, a differentiable method that integrates both graph topology and feature similarity as explicit regulation signals to enhance codebook utilization and diversity. Extensive experiments demonstrate that RGVQ significantly mitigates codebook collapse and improves the downstream performance, highlighting its broad applicability in learning expressive and transferable graph representations.

ETHICS STATEMENT

This work does not raise any obvious ethical concerns. All datasets used are publicly available and widely adopted in prior research.

REPRODUCIBILITY STATEMENT

We provide the implementation details and hyperparameters of the experiments in Appendix B to ensure reproducibility of our work. All datasets used in this paper are publicly available. Additionally, we provide the source code of our experiments in the supplementary material. Complete proofs of the theoretical analysis are included in Appendix A.

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

# APPENDIX

## A  PROOF

**Definition 1** (Minimum Codeword Radius). *Let $\mathbf{C} = \{e_i\}_{i=1}^{K}$ be the codewords in a codebook and $\mathrm{Vor}(e_i)$ be their Voronoi regions. The minimum codeword radius of the codebook is defined as*

$$\delta_c := \min_j \mathrm{dist}\big(c_j, \partial\mathrm{Vor}(c_j)\big), \tag{17}$$

*where $\partial\mathrm{Vor}(c_j)$ denotes the boundary of the Voronoi cell of $c_j$ and $\mathrm{dist}(\cdot, \cdot)$ is the Euclidean distance.*

**Definition 2** (Computation Trees (Chuang & Jegelka, 2022; Wang et al., 2024)). *Given a graph $\mathcal{G} = (\mathcal{V}, \mathcal{E})$, define $\mathcal{T}_v^L$ as the L-layer computation tree of node $v$ and $\mathcal{T}_v^1 = v$. $\mathcal{T}_v^L$ is constructed by recursively integrating the subtrees of neighborhoods of $v$.*

**Assumption 1** (Local consistency of Vector Quantization). *For two nodes $v_1$ and $v_2$ sampled uniformly at random from the graph, there exists a constant $\alpha \in (0, 1]$ such that*

$$\mathcal{P}(z_{v_1} = z_{v_2} \mid \|h_{v_1} - h_{v_2}\| \le \delta_c) \ge \alpha. \tag{18}$$

*Under this assumption, whenever two randomly sampled embeddings fall within the minimum codeword radius $\delta_c$, the probability that they are quantized to the same codeword is at least $\alpha$. Such local consistency guarantees that the quantizer does not behave arbitrarily for close nodes, which is a standard assumption used in K-Nearest-Neighbor clustering methods (Ding & He, 2004).*

We restate Theorem 1 in the main paper as below.

**Theorem 1** (Token Co-assignment Probability with GNN Encoder). *Given a L-layer GNN encoder $\phi$ with parameters $\mathbf{W} = (\mathbf{W_1}, \mathbf{W_2})$, for two nodes $v_1$ and $v_2$ sampled uniformly at random from the graph, the probability that two nodes are quantized to the same codeword $p := \mathbb{P}[z_{v_1} = z_{v_2}]$ satisfies the following lower bound:*

$$p \ge \alpha\left(1 - \frac{2\mathcal{B}_x}{\delta_c}\Big[C_1 + \sum_{\ell=1}^{L} C_2^\ell \mathbb{E}[D_\ell]\Big]\right), \tag{19}$$

*where $\alpha \in (0, 1]$ is the local quantization consistency constant, $\delta_c$ is the minimum codeword radius of the codebook, $C_1 = C_\sigma\mathcal{B}_{\mathbf{W}_1}$ and $C_2 = C_\sigma C_\rho C_g \mathcal{B}_{\mathbf{W}_2}$ are constants related to GNN parameters. Here $C_\sigma, C_\rho, C_g$, are Lipschitz term of GNNs. $\mathcal{B}_x$ is the bounded norm of node feature $x$, and $\mathcal{B}_{\mathbf{W}_1}, \mathcal{B}_{\mathbf{W}_2}$ denote bounded norm of $\mathbf{W}_1, \mathbf{W}_2$, respectively. $D_\ell = d_\ell d_{\ell-1}...d_1$, and $d_\ell$ indicates the number of children of the l-layer computation trees.*

*Proof.* Since the GNN encoder is based on message passing, each node's embedding depends on information aggregated from its local neighborhood. This process can be formalized as a computation tree $\mathcal{T}_v^L$ of depth $L$, rooted at node $v$, where messages are propagated from leaf nodes up to the root. The embedding $h_v$ can thus be viewed as a function of this tree, i.e., $h_v = \phi(\mathcal{T}_v^L)$. Let $x_v$ denote the node feature of $v$ and $\mathcal{N}_v$ be the set of its direct neighboring nodes, which correspond to the children of the computation tree $\mathcal{T}_v^L$. We begin by computing the embedding distance between two $L$-layer computation trees (Wang et al., 2024), whose embeddings are generated by the GNN $\phi$ with parameters $\mathbf{W} = (\mathbf{W}_1, \mathbf{W}_2)$, where $\mathbf{W}_1$ and $\mathbf{W}_2$ are the transformation of the root node and its neighboring nodes, respectively. Note that both $\mathbf{W}_1$ and $\mathbf{W}_2$ are Lipschitz continuous and are bounded by $\|\mathbf{W}_1\| < \mathcal{B}_{\mathbf{W}_1}$ and $\|\mathbf{W}_2\| < \mathcal{B}_{\mathbf{W}_2}$. For simplicity, we assume that the GNN parameters are shared across all layers, i.e., $\mathbf{W}$ remains the same at each layer, which does not affect the validity of the proof. Then the node embedding is computed as:

$$h_v = \phi(\mathcal{T}_v^L) = \sigma \left( \mathbf{W}_1 x_v + \mathbf{W}_2 \rho \left( \sum_{j \in \mathcal{N}(v)} g\left(\mathcal{T}_j^{L-1}(\mathbf{W})\right) \right) \right), \tag{20}$$

where $\sigma$ is the activation function, $\rho$ is the permutation-invariant aggregation, and $g$ is the update function in neural networks. For simplicity, we use $\mathcal{T}_v^L(\mathbf{W})$ to represent $\phi(\mathcal{T}_v^L)$.

Then, we have the upper bound of the distance between node embeddings:

$$\begin{aligned}
\Delta_{v_1,v_2}^L &= \left\| \mathcal{T}_{v_1}^L(\mathbf{W}) - \mathcal{T}_{v_2}^L(\mathbf{W}) \right\|_2 \\
&= \left\| \sigma\left(\mathbf{W}_1 x_{v_1} + \mathbf{W}_2 R(\mathcal{T}_{v_1})\right) - \sigma\left(\mathbf{W}_1 x_{v_2} + \mathbf{W}_2 R(\mathcal{T}_{v_2})\right) \right\|_2 \\
&\leq C_\sigma \left\| \mathbf{W}_1(x_{v_1} - x_{v_2}) + \mathbf{W}_2\left(R(\mathcal{T}_{v_1}) - R(\mathcal{T}_{v_2})\right) \right\|_2 \\
&\leq C_\sigma \left\| \mathbf{W}_1(x_{v_1} - x_{v_2}) \right\|_2 + C_\sigma \left\| \mathbf{W}_2\left(R(\mathcal{T}_{v_1}) - R(\mathcal{T}_{v_2})\right) \right\|_2 \\
&\leq C_\sigma \mathcal{B}_{\mathbf{W}_1} \left\| x_{v_1} - x_{v_2} \right\|_2 + C_\sigma \mathcal{B}_{\mathbf{W}_2} \left\| R(\mathcal{T}_{v_1}) - R(\mathcal{T}_{v_2}) \right\|_2,
\end{aligned} \tag{21}$$

where $R(\mathcal{T}_v) := \rho\left(\sum_{j \in \mathcal{N}(v)} g(\mathcal{T}_j^{L-1}(\mathbf{W}))\right)$. Considering the Lipschitz continuity of $\rho$ and $g$, then $\|R(\mathcal{T}_{v_1}) - R(\mathcal{T}_{v_2})\|_2$ is bounded as:

$$\begin{aligned}
\|R(\mathcal{T}_{v_1}) - R(\mathcal{T}_{v_2})\|_2 &\leq C_\rho \left\| \sum_{j \in \mathcal{N}(v)} g(\mathcal{T}_{v_1,j}^{L-1}) - \sum_{j \in \mathcal{N}(v)} g(\mathcal{T}_{v_2,j}^{L-1}) \right\|_2 \\
&\leq C_\rho \sum_{j \in \mathcal{N}(v)} \left\| g(\mathcal{T}_{v_1,j}^{L-1}) - g(\mathcal{T}_{v_2,j}^{L-1}) \right\|_2 \\
&\leq C_\rho C_g \sum_{j \in \mathcal{N}(v)} \left\| \mathcal{T}_{v_1,j}^{L-1} - \mathcal{T}_{v_2,j}^{L-1} \right\|_2 \\
&= C_\rho C_g \sum_{j \in \mathcal{N}(v)} \Delta_{v_1,v_2,j}^{L-1},
\end{aligned} \tag{22}$$

where $\Delta_{v_1,v_2,j}^{L-1} = \|\mathcal{T}_{v_1}^{L-1}(\mathbf{W}) - \mathcal{T}_{v_2}^{L-1}(\mathbf{W})\|_2$ is the distance of the embeddings between the $j$-th child of nodes $v_1$ and $v_2$. Here we assume that $v_1$ and $v_2$ have the same local structures by padding virtual nodes in their actual computation trees (Chuang & Jegelka, 2022), as shown in Figure 5, such that such that the $j$-th child of both $v_1$ and $v_2$ always exists. These virtual nodes do not have features and do not affect the actual computation and the generality of the proof. Let $d_l$ denote the number of branches, i.e., number of children at the $l-$th layer, the bound in Equation 22 can be simplified as:

$$\|R(\mathcal{T}_{v_1}) - R(\mathcal{T}_{v_2})\|_2 \leq C_\rho C_g d_{l-1} \max_{j \in \mathcal{N}(v)} \Delta_{v_1,v_2,j}^{L-1}. \tag{23}$$

This bound indicates that the most influential child of a node, i.e., a largely connected node will dominate all other children, and allows us to repeatedly expand $\Delta_{v_1,v_2,j}^{L-1}$ until the leaf nodes, where the distance is directly computed between input node features. Specifically, by recursively applying the above bound over tree depth $L$, we have:

$$\Delta_{v_1,v_2}^L \leq C_1 \|x_{v_1} - x_{v_2}\|_2 + C_2 d_{l-1} \max_{j \in \mathcal{N}(v)} \Delta_{v_1,v_2,j}^{L-1}, \tag{24}$$

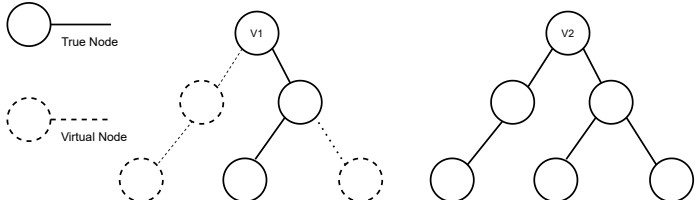

Figure 5: Adding virtual nodes to aligh the computation trees of $v_1$ and $v_2$.

where $C_1 = C_\sigma \mathcal{B}_{\mathbf{W}_1}$ and $C_2 = C_\sigma \mathcal{B}_{\mathbf{W}_2} C_\rho C_g$ are constants related to GNN parameters.

Assuming that the number of children at l-th layer does not exceed the maximum number of branches in the tree, such that $d_1, \ldots, d_L < d$, we have:

$$\Delta_{v_1, v_2}^L \leq C_1 \|x_{v_1} - x_{v_2}\|_2 + C_2 \sum_{j \in \mathcal{N}(v)} \Delta_{v_1, v_2, j}^{L-1}. \tag{25}$$

Given all node features $X$ are bounded by $\|X\|_2 \leq \mathcal{B}_x$ and the distance of $x_{v_1}$ and $x_{v_2}$ are bounded by $\|x_{v_1} - x_{v_2}\| \leq 2\mathcal{B}_x$. We further develop the bound as:

$$\Delta_{v_1, v_2}^L \leq 2\mathcal{B}_x \left( C_1 + \sum_{\ell=1}^L C_2^\ell D_\ell \right), \tag{26}$$

where $D_\ell = d_\ell d_{\ell-1} \cdots d_1$ denotes the total branching factor up to depth $\ell$. Equation 26 indicates that the distance between node embeddings are bounded by the feature difference and the degrees of nodes in the computation tree.

Then, we develop the lower bound of token co-assignment of uniformly sampled nodes $v_1$ and $v_2$. Let $z_{v_1}$ and $z_{v_2}$ be the quantized tokens assigned via codebook $\mathcal{C}$. For the lower bound, we consider the conservative condition for assigning to the same token. Then:

$$
\begin{aligned}
\mathbb{P}(z_{v_1} = z_{v_2}) &= \mathbb{P}(z_{v_1} = z_{v_2}, \|h_{v_1} - h_{v_2}\| \leq \delta_c) + \mathbb{P}(z_{v_1} = z_{v_2}, \|h_{v_1} - h_{v_2}\| > \delta_c) \quad (27) \\
&\geq \mathbb{P}(z_{v_1} = z_{v_2}, \|h_{v_1} - h_{v_2}\| \leq \delta_c) \\
&= \mathbb{P}(\|h_{v_1} - h_{v_2}\| \leq \delta_c) \cdot \mathbb{P}(z_{v_1} = z_{v_2} \mid \|h_{v_1} - h_{v_2}\| \leq \delta_c) \\
&\geq \mathbb{P}(\|h_{v_1} - h_{v_2}\| \leq \delta_c) \cdot \alpha.
\end{aligned}
$$

Using Markov's inequality on the non-negative random variable $\|h_{v_1} - h_{v_2}\|$ gives:

$$\mathbb{P}[\|h_{v_1} - h_{v_2}\| > \delta_c] \leq \frac{\mathbb{E}[\|h_{v_1} - h_{v_2}\|]}{\delta_c}. \tag{28}$$

Then, we derive lower bound of the probability that two nodes are assigned to the same token as:

$$\mathbb{P}[\|h_{v_1} - h_{v_2}\| \leq \delta_c] \geq 1 - \frac{\mathbb{E}[\|h_{v_1} - h_{v_2}\|]}{\delta_c} = 1 - \frac{\mathbb{E}[\Delta_{v_1, v_2}^L]}{\delta_c}. \tag{29}$$

According to Equation 26, the distance between node embeddings $\Delta_{v_1, v_2}^L$ has a upper bound, then its expectation are also bounded by:

$$\mathbb{E}[\Delta_{v_1, v_2}^L] \leq 2\mathcal{B}_x \left( C_1 + \sum_{\ell=1}^L C_2^\ell \mathbb{E}[D_\ell] \right) \tag{30}$$

Combining with Equation 29, we have:

$$\mathbb{P}[\|h_{v_1} - h_{v_2}\| \leq \delta_c] \geq \left( 1 - \frac{2\mathcal{B}_x}{\delta_c} \left[ C_1 + \sum_{\ell=1}^L C_2^\ell \mathbb{E}[D_\ell] \right] \right), \tag{31}$$

Substituting this lower bound into Equation 27, we got the lower bound of token co-assignment for uniformly sampled nodes:

$$\mathbb{P}(z_{v_1} = z_{v_2}) \geq \alpha \left( 1 - \frac{2\mathcal{B}_x}{\delta_c} \left[ C_1 + \sum_{\ell=1}^L C_2^\ell \mathbb{E}[D_\ell] \right] \right) \tag{32}$$

Table 5: Dataset statistics for selected datasets.

| Dataset | Domain | Task | # Graphs | Avg. #Nodes | Avg. #Edges | # Classes | Metric |
|---------|--------|------|----------|-------------|-------------|-----------|--------|
| CiteSeer | Citation | Node | 1 | 3,327 | 4,522 | 6 | Accuracy |
| Cora | Citation | Node | 1 | 2,708 | 10,556 | 7 | Accuracy |
| PubMed | Citation | Node | 1 | 19,717 | 88,651 | 3 | Accuracy |
| Computer | Co-purchase | Node | 1 | 13,752 | 491,722 | 10 | Accuracy |
| Photo | Co-purchase | Node | 1 | 7,650 | 238,163 | 8 | Accuracy |
| WikiCS | Web link | Node | 1 | 11,701 | 216,123 | 10 | Accuracy |
| Amazon-Ratings | Review | Node | 1 | 22,662 | 32,927 | 18 | Accuracy |
| Roman-Empire | Synthetic | Node | 1 | 24,492 | 93,050 | 5 | Accuracy |
| Questions | Synthetic | Node | 1 | 48,921 | 153,540 | 2 | ROC-AUC |
| FB15K237 | Knowledge | Link | 1 | 14,541 | 310,116 | 237 | Accuracy |
| WN18RR | Knowledge | Link | 1 | 40,943 | 93,003 | 11 | Accuracy |
| PCBA | Molecule | Graph | 437,929 | 26.0 | 28.1 | 128 | ROC-AUC |
| HIV | Molecule | Graph | 41,127 | 25.5 | 27.5 | 2 | ROC-AUC |

Note that this lower bound is a conservative estimate of the true co-assignment probability. This is because the bound only accounts for embedding pairs whose distance is within $\delta_c$. The actual probability may be higher. There may exist additional cases where $\|h_{v_1} - h_{v_2}\| > \delta_c$ but both embeddings still fall within the Voronoi region of the same codeword, such that $z_{v_1} = z_{v_2}$ still holds.

## B  EXPERIMENTAL SETUP

### B.1  DATASET

We use both homophilous and heterophilous graphs in our experiments. To implement empirical study and evaluate codebook utilization, we use various datasets, including Cora (Bojchevski & Günnemann, 2017), CiteSeer, PubMed (Namata et al., 2012), Amazon-Computer, Amazon-Photo (Shchur et al., 2018; McAuley et al., 2015), WikiCS (Mialon et al., 2021), Amazon-Ratings (Platonov et al., 2023), and Roman-Empire (Platonov et al., 2023). To assess transferability, we use cross-task and cross-domain datasets. Specifically, we use Cora, PubMed, and WikiCS for node classification; WN18RR (Shang et al., 2019) and FB15K237 Li et al. (2024b) for link prediction; and HIV (Hu et al., 2021) and PCBA (Chen et al., 2024) for graph classification. Finally, we evaluate serialization ability using the same datasets employed for codebook utilization. Detailed dataset statistics are summarized in Table 5.

### B.2  BASELINE

We use different baselines for the empirical study and three parts of our main experiments.

**Empirical Study and Codebook Utilization**. We primarily adopt codebook mitigation strategies originally developed in the vision and language domains, including EMA (Łańcucki et al., 2020), affine parameters (Huh et al., 2023), codebook reset (Zeghidour et al., 2021), and pretrained encoders (Zhao et al., 2024). We further include SimVQ (Zhu et al., 2024), which addresses the codebook collapse via one-single MLP layer over the latent basis vectors. Additionally, we compare with HQA-GAE (Zeng et al., 2025), a recent graph VQ model that applies a hierarchical VQ structure and an annealing strategy for codeword selection.

**Transferability**. To evaluate the effectiveness of RGVQ in learning transferrable graph tokens, we integrate it into a graph foundation model, i.e., GFT, and compare it with vanilla Graph VQ and its variants with different mitigation strategies. Moreover, we include supervised GNNs, i.e., GCN, GAT, and GIN, and graph self-supervised methods, i.e., DGI (Veličković et al., 2018), BGRL (Thakoor et al., 2021), GraphMAE (Hou et al., 2022), and GAINT (Chien et al., 2021). The supervised GNNs are trained directly on each target dataset, while the self-supervised methods and all GFT variants are pretrained on the full set of datasets and then fine-tuned per target.

Table 6: Hyperparameters of RGVQ for each dataset.

| Hyperparameter | Cora | Pubmed | Citeseer | Computer | Photo | WikiCS | Ratings | Roman | Questions |
|---|---|---|---|---|---|---|---|---|---|
| Hidden dimension | 256 | 256 | 256 | 256 | 256 | 256 | 256 | 256 | 256 |
| Learning rate | 0.001 | 0.001 | 0.001 | 0.001 | 0.001 | 0.001 | 0.001 | 0.001 | 0.001 |
| Weight decay | 1e-5 | 1e-5 | 1e-5 | 1e-5 | 1e-5 | 1e-5 | 1e-5 | 1e-5 | 1e-5 |
| Seed | 42 | 42 | 42 | 42 | 42 | 42 | 42 | 42 | 42 |
| Epochs | 1000 | 1000 | 1000 | 1000 | 1000 | 1000 | 1000 | 1000 | 1000 |
| Feature Reconstruction | 100 | 100 | 100 | 100 | 100 | 100 | 100 | 100 | 100 |
| Topology Reconstruction | 0.01 | 0.01 | 0.01 | 0.01 | 0.01 | 0.01 | 0.01 | 0.01 | 0.01 |
| $\beta$ | 1 | 1 | 1 | 1 | 1 | 1 | 1 | 1 | 1 |
| Temperature | 0.1 | 0.1 | 0.1 | 0.1 | 0.1 | 0.1 | 0.1 | 0.1 | 0.1 |
| Similarity function | Cosine | Cosine | Cosine | Cosine | Cosine | Cosine | Cosine | Cosine | Cosine |
| Top-$K$ | 20 | 20 | 20 | 20 | 20 | 20 | 20 | 20 | 20 |
| Sample number | 50 | 50 | 50 | 50 | 50 | 50 | 50 | 50 | 50 |
| GNN layers | 4 | 4 | 4 | 4 | 4 | 4 | 4 | 4 | 4 |

**Serialization**. To further evaluate the effectiveness of RGVQ in serialization, we integrate it into the Graph Quantized Transformer (GQT) (Wang et al., 2025), where discrete tokens serve as the input sequence to a vanilla Transformer backbone. We follow the original sequence reconstruction method and compare the performance of RGVQ-enhanced GQT against the original GQT, supervised GNNs, and graph transformers, including GraphGPS (Rampášek et al., 2022), SGFormer (Wu et al., 2023), Exphomer (Shirzad et al., 2023), and NodeFormer (Wu et al., 2022).

### B.3 IMPLEMENTATION DETAILS

**Empirical Study**. We provide the hyperparameters and experimental setup used in the empirical study of codebook perplexity. We jointly train the single-head VQ model and the GAT encoder using the link prediction and feature reconstruction tasks, along with the commitment loss and vocabulary loss. The task weights are set to 0.01, 100, 0.1, and 0.9, respectively. We train the model for 1000 epochs and report the highest perplexity during the training process for each method and a specific codebook size $K$. For all methods, we utilize the kmeans initialization and orthogonal regulation for the codebook, with a regularization weight of 0.1. The GNN consists of 4 layers with a hidden dimension of 256. AdamW is utilized as the optimizer with a learning rate of 1e-4 and a weight decay of 1e-5. For affine parameters, we use Euclidean distance and set the codebook decay to 0.9. For codebook reset, the threshold of deadcode is set to 10. For pretraining encoder, we pretrain the GNN encoder for 50 epochs before the joint training.

**Codebook Utilization**. The implementation of all collapse mitigation strategies is the same with the empirical study. For HQA-GAE, we use one-head VQ model and use the same hidden dimension and number of GNN layers as other methods, while all remaining hyperparameters follow the original paper. For RGVQ, we set the codebook size $K$ to 512. To construct the contrastive sample sets, for each node, we construct a pool of positive candidates by combining its 1-hop neighbors with the nodes that are most 20 similar in the input feature space (top-$K = 20$), which is measured by cosine similarity. From this pool, we sample 50 nodes with replacement as positive samples. The negative pool is defined symmetrically as all nodes that are neither neighbors nor feature positives, and we sample 50 negative nodes with replacement from the negative pool. The training weights for the link reconstruction, node feature reconstruction, contrastive regularization, commitment loss, and vocabulary loss are set to 0.01, 100, 1, 0.1, and 0.9, respectively. We set the temperature for the Gumbel-Softmax trick to 0.1. We train on each dataset for 1000 epochs to ensure convergence, and repeat the process 20 times to report the mean perplexity with standard deviations. Detailed hyperparameters for each dataset are summarized in Table 6.

**Transferability**. We use RGVQ as a plugin within the pretraining pipeline of GFT. Specifically, we retain the same pretraining tasks in GFT (Wang et al., 2024), including the link, node feature, and node embedding reconstruction tasks, and integrate RGVQ as a regularization term. Their weights are set to 100, 1, 0.01, and 10, respectively. For the backbone encoder, we utilize a 2-layer GCN model with ReLU activation, and set the codebook size to 512 and the hidden dimension to 256. We use AdamW optimizer with a learning rate of 1e-3 and weight decay of 1e-5. For data augmentation, we apply a link drop rate and the node-feature drop rate of 0.2. We pretrain the VQ tokens for 500

Table 7: Selected hyperparameters in GQT for each dataset.

| Dataset | GNN Encoder | | Quantizer | | Transformer | | | | |
|---|---|---|---|---|---|---|---|---|---|
| | # layers | # Hidden dim | # Codebooks | Codebook size | KNN | PPR | # Layers | # Heads | # FFN dim |
| Cora | 2 | 256 | 3 | 128 | 0 | 15 | 2 | 4 | 512 |
| CiteSeer | 2 | 256 | 3 | 128 | 5 | 15 | 2 | 4 | 512 |
| PubMed | 2 | 256 | 3 | 256 | 0 | 15 | 2 | 4 | 512 |
| Computer | 2 | 256 | 3 | 128 | 5 | 30 | 2 | 4 | 512 |
| Photo | 3 | 512 | 3 | 128 | 5 | 20 | 2 | 4 | 1024 |
| WikiCS | 2 | 256 | 3 | 128 | 5 | 30 | 2 | 4 | 512 |
| Amazon-Ratings | 4 | 512 | 3 | 128 | 5 | 20 | 2 | 4 | 1024 |
| Roman-Empire | 6 | 256 | 3 | 256 | 10 | 15 | 3 | 4 | 512 |
| Questions | 3 | 256 | 3 | 512 | 10 | 15 | 2 | 4 | 512 |

epochs on all datasets. During finetuning, we repeat each experiment 20 times to report the average performance with standard deviations. We finetune the model for 250 epochs using early stopping. For dataset splits, we follow the commonly used protocol for Cora and PubMed and utilize the predefined 10 splits with different seeds to report the downstream performance. Each split includes 20 labeled nodes per class for training. For WikiCS, we follow the recommended protocol by OGB and use the official split, reporting average performance across 20 splits (Mernyei & Cangea, 2020). For WN18RR, we utilize 86,835/3,034/3,134 links for training/validation/test, respectively. For FB15K237, we use 272,155/17,535/20,466 links for training/validation/test, respectively. For HIV and PCBA, we follow the official data split and utilize 80%/10%/10% for training/validation/test set (Hu et al., 2020).

**Serialization**. The training of GQT includes two parts: the VQ tokenizer and the backbone transformer. We detail the implementations and training hyperparameters below. For the VQ tokenizer, we follow the original paper and use Residual VQ (Wang et al., 2025). We retain all of the reconstruction tasks in the pretraining setting of GQT, including Deep Graph Infomax (DGI) (Veličković et al., 2018) and GraphMAE2 (Hou et al., 2023), and integrate RGVQ as a regularization term. For the tokenizer, we set the number of codebooks to three for GQT, GQT + EMA, GQT + AP, GQT + Reset, GQT + PT; and one for GQT + SimVQ, GQT + RGVQ. We choose codebook size from {128,256,512}. For the GNN encoder, we adopt GCN with ReLU activation, varying the number of layers from {2,3,4,6} and hidden dimensions from {256,512}. We pretrain the VQ tokenizer and the GNN encoder for 200 epochs until convergence. For training the vanilla transformer, we construct semantic links using K-Nearest-Neighbors, with K in {0,5,10}. To serialize the input graph sequence, we use Personalized PageRank (PPR) to generate a sequence for each node, with the sequence length selected from {15,20,30}. The transformer uses 2 or 3 layers, 4 attention heads, and a feedforward dimension of {512, 1024}. The detailed hyperparameters are summarized in Table 7. We train transformers with node labels together with the pretrained VQ tokenizer and GNN encoder, and report the average performance and standard deviations over 5 runs. For Cora, Pubmed, Citeseer, Computer, Photo, we follow the original settings in Wang et al. (2025), using 60%/20%/20% for training/validation/test. For WikiCS, we follow the predefined split in Mernyei & Cangea (2020) and report the average performance across 20 splits. For Amazon-Ratings, Roman-Empire, and Questions, we adopt the splits in Platonov et al. (2023), using 50%/25%/25% for training/validation/test, and report the mean performance over 10 random splits.

## C  ADDITIONAL EXPERIMENTS

**Empirical Study**. We additionally provide the quantization results on Cora and Citeseer datasets. The results shown in Figure 6 suggest that codebook collapse is a systematic problem in Cora and Citeseer datasets, even though the mitigation strategies are applied.

**Ablation Study**. We also provide the additional ablation study to further evaluate the contribution of each proposed module in RGVQ in this section. Here we use the normalized perplexity, defined as the ratio of utilized codebook entries to the total size of the codebook. First, we evaluate how codebook size affects the codebook utilization of RGVQ on more datasets. The results are shown in Figure 7(a). Across all datasets, RGVQ consistently maintains high normalized perplexity as the codebook size increases, showing strong robustness to the choice of codebook size. Notably, even with a large codebook size of 512, the model utilizes over 50% of the codebook capacity across

Table 8: Perplexity with varying number of GNN layers $L$.

| Dataset | $L = 1$ | | $L = 2$ | | $L = 3$ | | $L = 4$ | | $L = 5$ | |
|---|---|---|---|---|---|---|---|---|---|---|
| | VQ | RGVQ | VQ | RGVQ | VQ | RGVQ | VQ | RGVQ | VQ | RGVQ |
| Cora | 154.34 | 394.96 | 121.59 | 339.19 | 109.06 | 257.47 | 94.47 | 211.69 | 99.44 | 218.45 |
| Pubmed | 8.97 | 452.16 | 3.12 | 300.51 | 5.18 | 295.64 | 4.14 | 319.09 | 4.07 | 295.64 |
| Photo | 1.99 | 432.32 | 1.00 | 421.04 | 1.00 | 443.60 | 1.00 | 446.02 | 1.00 | 306.06 |
| Computer | 3.81 | 468.98 | 1.00 | 452.41 | 1.00 | 464.65 | 1.00 | 413.10 | 1.00 | 394.83 |
| Ratings | 32.64 | 414.10 | 15.59 | 295.28 | 10.80 | 213.42 | 13.29 | 200.93 | 9.14 | 207.18 |

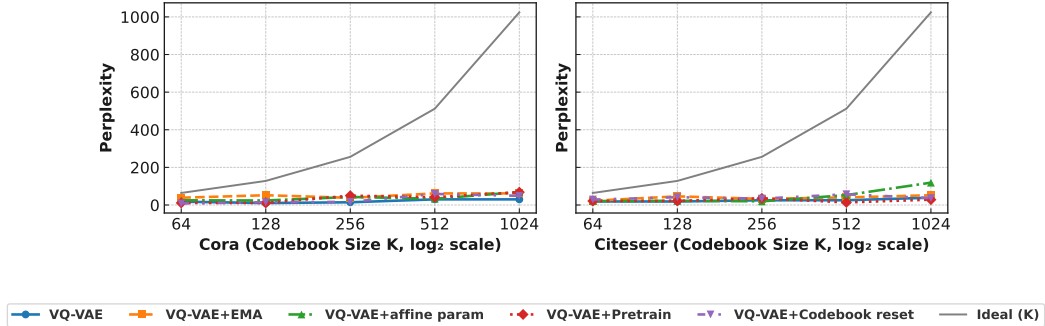

Figure 6: Codebook perplexites on graph datasets. The black lines indicate the optimal perplexities, i.e., codebook size K.

all datasets. Second, we investigate how the Gumbel–Softmax temperature $\tau$ affects normalized perplexity on more datasets. A lower temperature yields a codebook assignment distribution that is closer to one-hot. As shown in Figure 7(b), unlike some prior work (Zeng et al., 2025) that rely on temperature annealing, a relatively low and fixed temperature is sufficient to address the non-differentiability of deterministic VQ on selected datasets. Finally, we examine how the number of contrastive samples in structure-aware regularization affects normalized perplexity. As shown in Figure 7(c), even a small number of contrastive samples (e.g., $n = 10$ or $50$) achieves relatively high normalized perplexity, indicating effective codebook use. Increasing $n$ does not consistently lead to better utilization and may even degrade in some datasets, potentially due to added training noise.

**Converge Analysis**. We also provide the reconstruction loss and perplexity curves during the pre-training process of RGVQ and all baselines with codebook size $K = 512$. As shown in Figure 8, all baselines reach stable reconstruction loss within the first 250 epochs and remain stable afterwards, while they all collapse to less than 100 and do not recover. This confirms that the collapse is a problem for vanilla VQ and other anti-collapse solutions. Regarding reconstruction performance, collapse does not necessarily produce large reconstruction losses because a strong decoder can over-fit to node features or links even though usable tokens are limited. However, this phenomenon is fundamentally undesirable. When nodes collapse to the small portion of tokens, the discrete latent space becomes degenerate and ceases to reflect any structural or semantic diversity in the graph. In this situation, the VQ module fails to provide meaningful discrete representations. While the appropriate codebook size may depend on task-specific trade-offs between compression and expressiveness, our method offers a flexible and effective framework that preserves token diversity while scaling to larger codebook sizes.

**Influence of GNN Layer Number**. To better understand the relations between GNN layer number and quantization diversity, we evaluate how the number of GNN layer $L$ affects the quantization perplexity in RGVQ and vanilla VQ. Based on the results in Table 8, we make the following observations: (1) As the layer number $L$ increases, the perplexity of vanilla VQ consistently decreases. Deeper GNNs suffer from over-smoothing, causing node representations to fall more easily within the radius of the same codeword, resulting in less diverse quantization. (2) RGVQ is robust for different layers because it provides explicit regularization. (3) These empirical trends are fully aligned with Theorem 1, which relates token co-assignment probabilities to the computation-tree depth $L$.

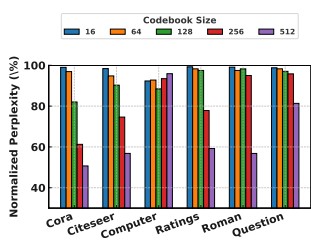 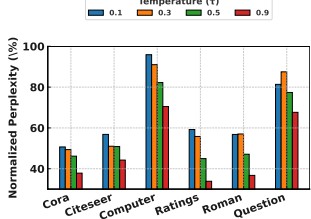 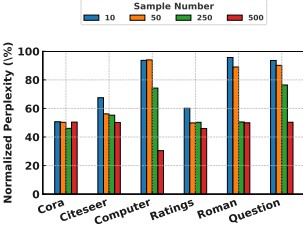

(a) Codebook size vs. Perplexity.    (b) Temperature ($\tau$) vs. Perplexity.    (c) Sample Number vs. Perplexity.

Figure 7: Ablation study results.

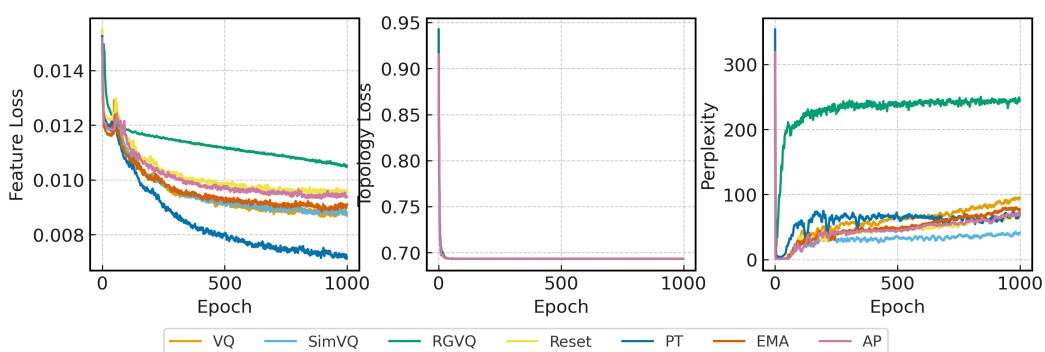

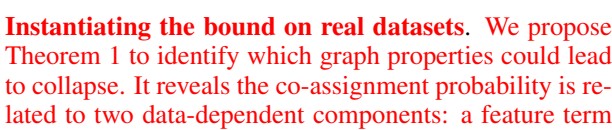

Figure 8: Reconstruction loss and perplexity during the pretraining process on Cora

**Influence of Codebook Diversity**. To better understand the relations between quantized diversity and the downstream performance, we evaluate how perplexity affects the node classification results. We select different pretraining checkpoints of RGVQ to reflect different perplexities. At the downstream stage, we utilize the pretrained tokens and finetune with node labels. The results are shown in Figure 9. We observe a consistent positive correlation between perplexity and accuracy across datasets. This suggests that a more diverse codebook can capture finer-grained structural patterns, enabling the model to learn more discriminative embeddings.

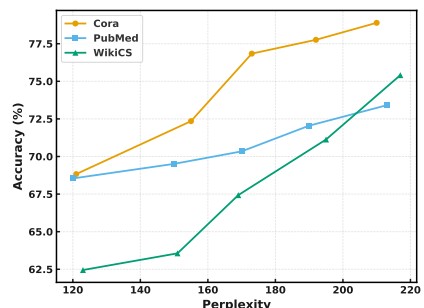

Figure 9: Correlation between graph downstream performance and codebook perplexity.

**Instantiating the bound on real datasets**. We propose Theorem 1 to identify which graph properties could lead to collapse. It reveals the co-assignment probability is related to two data-dependent components: a feature term $\mathcal{B}_x$ involving the feature norm, and a structure term involving the computation-tree expansion $\mathbb{E}[D_l]$. Other constants, such as $C_1$ and $C_2^\ell$, are model-dependent Lipschitz constants that cannot be estimated reliably. Thus, we use PCA@95 as a proxy for effective feature variation and the average node degree as a proxy for the expectation of the degrees of computation trees to instantiate the bound on real graphs. From the Table 9, we observe that across all 8 datasets, these characteristics align with the quantization results of vanilla VQ: datasets with high average degree (Photo, Computer) or low PCA@95 (Ratings, Roman, Questions) exhibit lower perplexity, whereas datasets with both higher PCA@95 and lower degree (Cora, Citeseer) exhibit much weaker collapse. This supports our theoretical insight that feature redundancy and structural redundancy drive collapse in Graph VQ. It should be noted that we do not attempt to numerically separate the two terms in the bound, nor do we use it to predict the collapse.

Table 9: Dataset statistics (PCA@95, average degree) and measured codebook perplexity across 8 graph datasets.

| Dataset | Cora | Pubmed | Citeseer | Photo | Computer | Ratings | Roman | Questions |
|---|---|---|---|---|---|---|---|---|
| **PCA@95** | 802 | 410 | 1459 | 611 | 646 | 194 | 141 | 160 |
| **Avg Degree** | 4.90 | 5.50 | 3.74 | 32.13 | 36.76 | 8.60 | 3.91 | 7.28 |
| **Perplexity** | 94.47 | 4.14 | 60.09 | 1.00 | 1.00 | 13.29 | 10.84 | 20.78 |

## D  COMPLEXITY ANALYSIS

Assume a $L$-layer GNN, a codebook of size $K$, and hidden dimension of $d$, the number of nodes and links are denoted as $|\mathcal{V}|$ and $|\mathcal{E}|$ respectively. We divide the complexity analysis into two parts: Pre-computation of contrastive set and quantization process.

**Pre-computation of Contrastive Set**. Before training, RGVQ constructs for each node sets of positive and negative samples, based on both structural and feature similarity. This step is performed once and reused during training. To implement this, neighbors are first extracted by scanning the adjacency matrix, which requires $O(|\mathcal{E}|)$ time. Then compute feature distances between each node and all others will take $O(|\mathcal{V}|^2 d)$ time. However, in practice, we adopt a sampling strategy: for each node, we sample $M$ non-neighbor nodes (where $M$ is a small constant, e.g., 100) and compute their feature similarity. This limits the total cost of semantic similarity computation to $O(|\mathcal{V}| \cdot M \cdot d)$, which is linear in the number of nodes. After collecting both structurally and semantically similar candidates, we perform top-$k$ selection for each node to finalize its positive sample set, costing $O(|\mathcal{V}| \log k)$ time in total. Negative pairs are sampled from the set of all nodes excluding the positives. Thus, the overall time complexity of the contrastive set construction process is $O(|\mathcal{E}| + |\mathcal{V}| \cdot M \cdot d + |\mathcal{V}| \log k)$, which is linear in the number of nodes and edges under fixed $M$ and $k$.

**Quantization Process**. The time and space complexity of the GNN encoder are $O(Ld^2|\mathcal{V}| + Ld|\mathcal{E}|)$ and $O(Ld^2 + Ld|\mathcal{V}| + |\mathcal{E}|)$, respectively. The decoder has the same complexity. RGVQ computes the distance between each node embedding and all $K$ codewords, and estimates the soft assignment distribution via the Gumbel-Softmax trick. This process requires $O(|\mathcal{V}|Kd)$ time and $O(|\mathcal{V}|K)$ space. Finally, RGVQ regularizes the assignment distributions using the InfoNCE loss between node pairs. For each node, this involves computing similarities with $k$ positive and $k$ negative samples, each over $K$-dimensional distributions. The total cost is $O(|\mathcal{V}| \cdot (2k) \cdot K)$.

## THE USE OF LLMS

We used OpenAI's ChatGPT-5 to assist with code for visualization in this paper. Specifically, based on the experimental data, the tool was used to improve the visualization code in order to enhance the readability and overall quality of the figures. We also used ChatGPT-5 to improve the grammar, clarity, and language quality of the manuscript, in a manner comparable to standard writing-assistance tools.

