# OpenReview forum: "Graph is a Natural Regularization: Revisiting Vector Quantization for Graph Representation Learning"
_ICLR.cc/2026/Conference — Submitted to ICLR 2026_

### Official Review · Reviewer_8u7x · 2025-10-16

**Soundness:** 3
**Presentation:** 3
**Contribution:** 4
**Rating:** 6
**Confidence:** 3

**Summary:**

This paper investigates why vector quantization (VQ) methods often fail when applied to graph representation learning, exhibiting severe *codebook collapse*, where most tokens remain unused.
The authors identify two intrinsic causes:
(1) graph-structured data exhibit strong feature redundancy and non-i.i.d. sampling, which biases early token assignments, and
(2) the standard VQ training objective reinforces this imbalance via a self-amplifying loop.

To address this, the paper proposes Regularized Graph Vector Quantization (RGVQ).
The method introduces (i) a Gumbel-Softmax reparameterization to allow differentiable soft token assignment, ensuring all codewords receive gradient updates, and (ii) a structure-aware regularization that leverages graph topology and feature similarity to encourage balanced token utilization.
Extensive experiments on standard benchmarks demonstrate that RGVQ can improve codebook utilization and downstream graph learning performance over existing Graph-VQ methods.

Overall, the paper offers a clear theoretical analysis of why codebook collapse arises in graph domains and provides an elegant, empirically supported remedy.

**Strengths:**

* **Clear motivation and theoretical insight:** The authors provide a compelling and formal explanation of why VQ collapses more severely on graphs than on i.i.d. data, linking structural redundancy and non-independent sampling to early token imbalance.
* **Strong empirical validation:** RGVQ consistently improves codebook perplexity and downstream performance across multiple benchmarks and graph architectures.
* **General applicability:** The approach is modular and can be integrated into various Graph-VQ frameworks.
* **Theoretical–empirical alignment:** The analysis of token-usage distribution supports the proposed mechanism and the effectiveness of the regularization.

**Weaknesses:**

* **Scalability limitations:** The experiments are conducted on medium-sized graphs (e.g., Cora, PubMed, Amazon). It is unclear whether RGVQ can scale to very large graphs or industrial settings where full pairwise structural regularization becomes costly.
* **Limited baselines:** Although several Graph-VQ methods are compared, broader baselines such as tokenization-based GNN compression or self-supervised graph representation models could provide stronger context.
* **Ablation depth:** The paper demonstrates the importance of each component, but further analysis (e.g., computational overhead, impact of temperature τ on performance, not only perplexity) would strengthen the empirical claims.
* **Generality of the “natural regularization” claim:** The idea that “GRAPH ISANATURALREGULARIZATION” is conceptually appealing, but its formal definition and theoretical boundary could be discussed more precisely.
* The appendix includes a useful computational complexity analysis for the structure-aware regularization, which helps clarify scalability concerns. However, the current appendix hyperlinks seem broken or unreferenced, making it difficult for readers to locate those details. It would be helpful to fix cross-references and explicitly summarize the main complexity results in the main text for better readability.

**Questions:**

1. Have the authors considered approximating the structure-aware regularization through local sampling or hierarchical neighborhood aggregation to improve scalability on very large graphs?
3. Could the proposed RGVQ framework be extended to heterogeneous or dynamic graphs where structural relations evolve over time?
4. The Gumbel-Softmax temperature τ appears crucial to token utilization—how sensitive is model performance to τ, and can it be adaptively learned instead of tuned manually?
5. Would integrating RGVQ into large Graph Foundation Models (e.g., GROVER, GraphMAE) yield consistent benefits, or are there observed limitations when scaling up?

---

> ### Author Response · Authors · 2025-11-24
> **Response to the Weaknesses and Questions Raised by Reviewer 8u7x**
>
> We sincerely thank the reviewer for recognizing the significance and contributions of our work. Below, we address the raised weaknesses and questions in detail.
>
> ### Respond to W1 and Q1: Scalability
> We provide the details about the scalability and complexity analysis in **General Response Part 5** and **Appendix D**.
>
> Specifically, for each node, to avoid similarity computation over the entire graph, we adopt a sampling strategy: each node samples only $M$ non-neighbors (e.g., $M=100$) and computes similarity within this subset. This reduces the semantic-similarity cost to $O(|\mathcal{V}| \cdot M \cdot d)$, which is linear in the number of nodes. These feature-similar nodes and neighbors of the node form the positive pool.
>
> From the positive pool, we randomly sample 50 positives with replacement. Negatives are sampled from nodes that are neither neighbors nor feature-similar.
>
> To scale to large graphs, we apply the same sampling procedure for every input mini-batch, where positive/negative sets are dynamically constructed per batch. We additionally report scalability results on large graphs in **General Response Part 5**, where RGVQ consistently remains efficient and competitive.
>
> ### Response to W3 and Q3: More ablation studies
> We thank the reviewer for pointing out the valuable feedback. Here we added additional an ablation study to assess the downstream performance of RGVQ+GFT with different temperature. Additionally, we added  Drop-one ablation in the revised paper, please refer to **General Response Part 4** for more details.
>
> | Dataset | Metric      | 0.9                 | 0.7                  | 0.5                  | 0.3                  | 0.1                  |
> |---------|-------------|----------------------|-----------------------|-----------------------|-----------------------|-----------------------|
> | **Cora**    | Perplexity | 121.05 ± 10.98       | 189.64 ± 6.27         | 195.05 ± 7.36         | 203.03 ± 8.64         | 211.69 ± 5.27         |
> |         | Acc         | 79.01 ± 0.56         | 80.32 ± 0.39          | 80.38 ± 0.44          | 80.57 ± 0.37          | 80.85 ± 0.73          |
> | **PubMed**  | Perplexity | 198.69 ± 10.54       | 210.03 ± 9.87         | 243.45 ± 9.38         | 273.38 ± 13.19        | 319.09 ± 10.40        |
> |         | Acc         | 76.24 ± 1.21         | 76.39 ± 1.16          | 76.51 ± 1.09          | 77.09 ± 1.23          | 77.46 ± 0.94          |
> | **WikiCS**  | Perplexity | 123.35 ± 5.07        | 151.49 ± 3.89         | 169.74 ± 4.57         | 183.35 ± 3.62         | 228.82 ± 5.96         |
> |         | Acc         | 79.65 ± 0.93         | 79.88 ± 0.56          | 79.96 ± 0.65          | 79.95 ± 0.44          | 80.01 ± 0.52          |
>
> From the table, we observe a clear trend: higher perplexity consistently correlates with better node classification accuracy across all datasets. This further supports our claim that avoiding early token collapse leads to stronger representations. In addition, the results show that low temperatures yield strong performance without requiring any annealing strategies.
>
> Regarding computation overhead, according to the complexity analysis in **General Response Part 5**, we can easily conclude that temperature does not influence the computation overhead, while the complexity of the construction of contrastive sets and InfoNCE regularization are both linear to the number of contrastive samples.
>
> ### Response to W4: Definition of regularization and theoretical boundary
>
> In this paper, we formally show that the unique characteristics of graphs are closely related to codebook collapse. Specifically, our empirical study and theoretical analysis identify that both feature redundancy and structural redundancy correlate with biased token assignments. Based on these insights, we explicitly regularize token assignment distributions using feature similarity and graph connectivity. In this sense, we refer to “graph as a natural regularization.” For deeper analysis of the theoretical boundary, please refer to **General Response Part 3**.
>
> ### Response to W5: Hyperlinks.
> We thank the reviewer for pointing out this problem. We have checked all the hyperlinks and they should work in the revised version.
>
>
> ### Response to Q2: Extension to heterogeneous or dynamic graphs
> Yes. Our framework can be naturally and easily extended to heterogeneous and dynamic graphs because both key components (Gumbel-Softmax reparameterization and structure-aware regularization) are GNN-agnostic and only require node embeddings and the computation of feature similarity. VQ-based tokenization beyond homogeneous static graphs is promising, and RGVQ can be readily adapted to mitigate collapse problems.

---

> ### Author Response · Authors · 2025-11-24
>
> ### Response to W2 and Q4: More baselines, integrating RGVQ in othe GFMs
>
> Our method is designed to specifically address the token learning problem in Graph VQ models, where codebook collapse is a fundamental bottleneck. RGVQ directly serves as a plug-in within the VQ framework.
>
> Integrating VQ tokens into other GFMs (e.g., GROVER, GraphMAE) is a promising direction, but these models do not currently rely on VQ. Extending RGVQ to such architectures would require adapting or introducing VQ-based token modules, which we consider an important research direction for future work. Moreover, we will add more baselines in the camera-ready version.

---

> > ### Comment · Reviewer_8u7x · 2025-11-25
> > **Thank the authors for the reply.**
> >
> > The authors provide a positive and detailed rebuttal, I therefore maintain my original score.

---

> > > ### Author Response · Authors · 2025-11-25
> > >
> > > Thank you again for your time and valuable insights.

---

### Official Review · Reviewer_JNRk · 2025-10-28

**Soundness:** 2
**Presentation:** 2
**Contribution:** 2
**Rating:** 4
**Confidence:** 4

**Summary:**

This paper looks at codebook collapse in graph vector quantization. The authors argue that when quantize node embeddings into a discrete codebook, only a tiny handful of codewords actually get used, and this is especially bad on graphs since many nodes look locally redundant. They claim this hurts any attempt to build reusable “graph tokens” for transfer or for Transformer-style models on graphs. To fix it, they propose RGVQ and claim that it can both increases codebook perplexity (so the dictionary is actually used) and improve downstream accuracy when plugged into recent graph-token pipelines like GFT and GQT. After reviewing this paper,  I give a **weak reject rating (4) since some core pieces of the training recipe, baselines, and theoretical link are not yet fully convincing.**

**Strengths:**

1. Problem focus is well motivated. Treating collapse itself as the main object of study (instead of just reporting final accuracy) feels important for the emerging “graph as tokens” paradigm. The paper makes a decent case that collapse is systematic in graph VQ, not just an odd failure case.

2.  A simple, generally pluggable fix. RGVQ is made of two ideas that our community already understand (Gumbel-Softmax instead of hard argmax, plus a structure-aware contrastive regularizer), and the paper shows that dropping this module into GFT and GQT not only boosts codebook usage but also leads to better transfer accuracy on standard graph benchmarks. I think that practicality is attractive if it really generalizes.

**Weaknesses:**

1. **Theory-to-method gap.** The theoretical part (Thm 1) gives a lower bound saying that nodes with similar features and local computation trees are very likely to get mapped to the same codeword under standard VQ. This motivates why graphs collapse. But the constants are not instantiated in a way that proves the bound is non-vacuous on real data, and the theorem analyzes hard VQ while the proposed method uses Gumbel-Softmax and a contrastive regularizer. There is no formal argument that these two ingredients actually break the self-reinforcing loop identified by the theorem. Right now the theory reads more like intuition than a guarantee.

2. **Ambiguity in the actual training loss.** The paper first defines a soft assignment distribution $\tilde{p}_i$ and the soft quantized embedding $\tilde{z}_i = \sum_j \tilde{p}_i(e_j|h_i) e_j$, but later reuses the classic VQ loss with codebook and commitment terms between $h$ and $z$. It is not stated clearly whether $z$ there is $\tilde{z}$, a sampled hard codeword, or something else, nor where stop-gradients are applied. Since the main claim is “inactive codewords finally get gradients,” I think the paper needs a precise forward/backward description or pseudocode. Otherwise reproduction (and trust in the mechanism) is shaky.

3. **Structure-aware contrastive term underspecified.** The method builds a positive set (neighbors or top-$k$ most similar nodes) and a negative set (non-neighbors and not in top-$k$), then uses an InfoNCE-style loss over assignment distributions. However, details like how $k$ is chosen, how similarities are computed, how negatives are sampled, and how this scales beyond citation-size graphs are not fully spelled out in the main paper and appendix. The claim that RGVQ is an easy drop-in module would be much stronger if these knobs were made explicit and justified.

4. **Baseline fairness concerns.** The paper reports that plain Graph VQ and multiple known anti-collapse tricks can end up with perplexity near 1.0 even with a 512-codeword dictionary. If that is true, reconstruction should be terrible because essentially everyone maps to the same codeword. I would like to see evidence that these baselines were genuinely trained to convergence, not just stuck in a bad local optimum or stopped early. Similarly, for downstream tasks we only see “GFT vs GFT+RGVQ” and “GQT vs GQT+RGVQ,” but not “GFT + EMA/codebook reset/etc.” or “GQT + SimVQ.” Without those ablations, it is hard to isolate how much of the reported gain is specific to solving collapse versus just adding one more regularizer.

5. **Scalability and clarity.** The paper positions this as relevant for large-scale graph foundation modeling, but experiments are still mostly on medium-size benchmarks (citation graphs, Amazon co-purchase graphs, WikiCS, Roman-Empire). There is no quantitative discussion of memory or runtime overhead from doing structure-aware positive/negative sampling, which in the naive form sounds at least quadratic if you keep nearest-neighbor sets globally. This weakens the generality claim.

**Questions:**

1. In the final loss $L_{\text{VQ}}$, do you backprop through $\tilde{z}_i$ (the soft combination of all codewords via Gumbel-Softmax), or through a hard assignment $z_i$? Please give exact forward/backward steps, including where gradients are stopped. A short pseudocode block would help.

2. Can you report “GFT + EMA,” “GFT + codebook reset,” “GQT + SimVQ,” etc., not just vanilla vs RGVQ? Otherwise it is impossible to tell whether RGVQ is uniquely effective for collapse or just another regularizer that any strong baseline could also benefit from.

3. In Table 1, where vanilla Graph VQ shows perplexity $\approx 1.0$ for a 512-size codebook, what are the reconstruction losses and downstream accuracies at that point? Are these runs actually converged? If they are already unusably bad, then the gap to RGVQ might just reflect that you fixed a degenerate run, not that you improved an actually competitive baseline.

4. Them 1 is used to argue that structural redundancy in graphs drives collapse. Can you make that bound non-vacuous by instantiating the constants on real datasets, and then explain concretely how Gumbel-Softmax and the structure-aware contrastive term attack the mechanism identified by the theorem?

5. How does the positive/negative sampling scale on large graphs? Do you approximate nearest neighbors, cache neighborhoods, or subsample batches? Please quantify runtime and memory overhead relative to plain Graph VQ, since you claim RGVQ as broadly applicable to “graph foundation models.”

---

> ### Author Response · Authors · 2025-11-24
> **Response to the Weaknesses Raised by Reviewer JNRk**
>
> We would like to thank the reviewer for their constructive feedback. Please find our responses below:
>
> ### Respond to W1 and Q4: Theory-to-method gap.
>
> (i) Please see the General Response **Part 3** for detailed discussion about instantiating Theorem 1 on real datasets.
>
> (ii) Breaking the VQ dynamics: The self-loop dynamics of deterministic VQ comes from the hard assignment: if a codeword $e_k$ is never selected by the hard assignment, then its gradient is identically zero and is more likely to be discarded in the subsequent training process. RGVQ mainly break this loop by ensuring gradients to all codewords via Gumbel-softmax and contrastive regularization, and we explain the gradient flow here:
>
> **Gradients from regularization term**
>
> For the regularization loss $\mathcal L_{\mathrm{reg}}$, the gradient w.r.t. a codeword $e_k$ flows through the chain $
> L_{\mathrm{reg}} \rightarrow \tilde p \rightarrow p \rightarrow \pi _i \rightarrow e_k,
> $ which gives:
>
> $\frac{\partial \mathcal L_{\mathrm{reg}}}{\partial e_k}=\frac{\partial \mathcal L_{\mathrm{reg}}}{\partial \tilde p_i(e_k \mid h_i)}
> \frac{\partial \tilde p_i(e_k \mid h_i)}{\partial p_i(e_k \mid h_i)}
> \frac{\partial p_i(e_k \mid h_i)}{\partial \pi _i}
> \frac{\partial \pi _i}{\partial e_k},$
>
> where $\pi_i = -||h_i - e_k\||$,
> $p_i(\cdot \mid h_i)=\mathrm{Softmax}(\pi)$, and
> $\tilde p_i(\cdot \mid h_i)=\mathrm{Softmax}_\tau(\log p_i(\cdot\mid h_i)+g).$
>
> For softmax operators, we have:
>
> $\frac{\partial \tilde p_i(e_k\mid h_i)}{\partial p_i(e_k\mid h_i)}
> \neq 0,\quad
> \frac{\partial p_i(e_k\mid h_i)}{\partial \pi}\neq 0
> \quad, \forall k.$
>
> Moreover, $\pi$ depends smoothly on $e_k$ with
>
> $\frac{\partial \pi}{\partial e_k} = -2(h_i-e_k)\neq 0.$
>
> Therefore, every codeword $e_k$ receives gradients from the regularization term.
>
> **Gradients from Vocabulary Loss**
>
> As we utilize soft assignment in the forward pass, i.e., $\tilde{z}_i = \sum_j \tilde p_i(e_j|h_i)e_j$, the gradients from vocabulary loss $||\text{sg}[\mathbf H] - \mathbf{ \tilde Z}\||$ in Equation 16 are also passed to every codeword.
>
> Since the stop-gradient operator blocks gradients to $H$, we have
>
> $\frac{\partial \mathcal L_{\mathrm{codebook}}}{\partial e_k} = 2 ||\tilde z_i-\mathrm{sg}[h_i]|| \frac{\partial \tilde z_i}{\partial e_k}.$
>
> The derivative of $\tilde z_i$ w.r.t.\ $e_k$ can be fomalized as:
>
> $\frac{\partial \tilde z_i}{\partial e_k} = \tilde p_i(e_k\mid h_i) + e_k \frac{\partial \tilde p_i(e_k\mid h_i)}{\partial e_k}\quad
> $
>
> Based on the derivation of gradient from regularization, we have
>
> $\frac{\partial \tilde p_i(e_k\mid h_i)}{\partial e_k}\quad \neq 0, \quad \forall k.$
>
> Therefore, every codeword $e_k$ receives gradients from the codebook loss.
>
> ### Respond to W2 and Q1: Ambiguity in the actual training loss.
> We sincerely thank the reviewer for this constructive feedback. We substitute soft quantized embedding $\mathbf{\tilde z}$ in the reconstruction loss, commitment loss, and vocabulary loss in Equation 16. And we use the derived assignment probability $\tilde p$ to compute the regularization loss. We have added a pseudocode of the training process in the main text, and revised the actual training loss (Equation 16) for better readability.

---

> ### Author Response · Authors · 2025-11-24
>
> ### Respond to W3, W5 and Q5: Structure-aware contrastive term underspecified and scalability
>
> We sincerely thank the reviewer for raising this important question.
>
> **Specify the implementation of contrastive term**:
>
> We provide detailed complexity analysis and results on large graphs in **General Response Part 5**. Specifically, for each node, we first construct a pool of positive candidates for each node by combining its 1-hop neighbors with the $\text{top-k}=20$ most similar nodes in the input feature space, measured by **cosine similarity** (We use this setting for all experiments in this paper). To compute the feature similarites between nodes, we adopt a sampling strategy to avoid quadratic complexity: for each node, we sample $M$ non-neighbor nodes (where $M$ is a small constant, e.g., 100) and compute their feature similarity. This limits the total cost of semantic similarity computation to $O(|\mathcal{V}| \cdot M \cdot d)$, which is linear in the number of nodes. From the positive pool, we randomly sample 50 nodes with replacement as positive samples (We vary the number of samples in the ablation study). The negative pool consists of nodes that are neither neighbors nor feature positives, from which we sample 50 negatives with replacement. For the complexity analysis of other component, please refer to **General Response Part 5** for more details.
>
> **Scalability of contrastive term**:
>
> Our model is easy to scale to large graphs, as we adopt the same strategy for each minibatch. For example, we implement the same sampling strategy and construct the contrastive set in each input mini-batch. We additionally include the performance of RGVQ against vanilla VQ on ogbn-arxiv, ogbn-product, and ogbn-proteins, as shown below:
>
> | Method | ogbn-arxiv        | ogbn-products      | ogbn-proteins       |
> |--------|--------------------|---------------------|----------------------|
> | VQ     | 6.43 ± 0.94       | 10.13 ± 1.05       | 9.85 ± 0.86          |
> | RGVQ   | 305.13 ± 20.3     | 223.47 ± 10.81     | 254.87 ± 0.67        |

---

> ### Author Response · Authors · 2025-11-24
>
> ### Respond to W4 and Q2, Q3: Baseline fairness concerns.
>
> We thank the reviewer for this suggestion.
>
> **More baselines**:
>
> We added more baselines in the experiments, including GFT with anti-collapse methods (Table 2 in the revised paper) and GQT with anti-collapse methods (Table 3 in the revised paper).
> | Method         | Cora | PubMed | WikiCS | WN18RR | FB15K237 | HIV | PCBA | Avg. |
> |----------------|------|--------|--------|--------|----------|-----|------|------|
> | GFT            | 78.35±1.07 | 73.39±1.68 | 79.13±0.32 | 90.87±0.25 | 89.89±0.27 | 72.16±1.69 | 72.74±1.23 | 79.50 |
> | GFT + EMA      | 79.44±0.89 | 74.01±1.57 | 78.94±0.41 | 90.58±0.43 | 89.75±0.19 | 72.39±1.52 | 73.04±1.01 | 79.73 |
> | GFT + AP       | 79.69±1.07 | 75.05±0.86 | 79.73±0.35 | 89.56±0.18 | 89.05±0.18 | 71.86±1.53 | 71.48±0.99 | 79.48 |
> | GFT + Reset    | 80.07±0.91 | 75.51±0.69 | 79.85±0.33 | 91.18±0.43 | 88.09±0.23 | 72.79±1.65 | 71.95±0.85 | 79.92 |
> | GFT + PT       | 78.57±0.86 | 74.12±1.05 | 72.75±1.72 | 88.63±0.15 | 88.45±0.17 | 71.01±1.74 | 73.73±1.12 | 78.18 |
> | GFT + SimVQ    | 77.61±0.73 | 76.41±1.28 | 76.57±0.68 | 82.72±0.53 | 82.03±0.35 | 66.57±1.35 | 69.90±0.91 | 75.97 |
> | **GFT + RGVQ** | **80.85±0.73** | **77.46±0.94** | **80.10±0.52** | **91.32±0.26** | **90.45±0.31** | **74.10±1.49** | **75.68±0.99** | **81.42** |
>
> | Method        | Cora | PubMed | Citeseer | Photo | Computer | WikiCS | Ratings | Roman | Questions |
> |---------------|------|--------|----------|--------|----------|---------|---------|--------|-----------|
> | GQT           | 86.44±1.58 | 81.60±1.35 | 73.14±1.26 | 94.46±0.68 | 92.13±0.23 | 80.03±0.19 | 54.04±0.12 | 89.85±0.73 | 76.52±1.52 |
> | GQT + EMA     | 86.23±1.19 | 81.41±1.24 | 73.08±1.58 | 94.01±0.57 | 91.95±0.18 | 79.98±0.23 | 54.10±0.08 | 89.91±0.51 | 75.94±1.16 |
> | GQT + AP      | 85.89±0.94 | 83.31±0.97 | 72.56±1.38 | 96.15±1.21 | 94.46±0.36 | 82.03±0.59 | 54.54±0.24 | 90.46±0.52 | 76.96±1.17 |
> | GQT + Reset   | 86.15±1.07 | 83.50±1.01 | 71.59±1.37 | 95.15±0.55 | 94.79±0.48 | 82.84±0.23 | 54.41±0.17 | 90.50±0.42 | 78.13±0.98 |
> | GQT + PT      | 85.71±1.44 | 80.92±1.15 | 79.53±1.23 | 94.74±0.76 | 92.35±0.35 | 75.65±0.78 | 54.50±0.14 | 89.76±0.68 | 76.74±1.34 |
> | GQT + SimVQ   | 86.02±1.64 | 82.56±1.02 | 72.58±1.14 | 95.21±0.77 | 94.23±0.21 | 81.78±0.32 | 53.98±0.15 | 90.15±0.66 | 76.35±1.21 |
> | **GQT + RGVQ** | **88.34±1.32** | **86.54±1.41** | **81.25±1.01** | **97.66±1.05** | **95.67±0.36** | **83.58±0.66** | **55.16±0.19** | **90.98±0.66** | **78.26±1.07** |
>
> From these tables, we make the following observations: Since downstream performance is often bottlenecked by representation quality, RGVQ not only ensures better codebook utilization during pre-training but also yields consistent improvements on downstream tasks. Moreover, it should be noted that the downstream performance largely depends on the backbone architecture (e.g., GQT’s Residual VQ) and the supervision from node labels during fine-tuning. Therefore, even when the codebook collapses, some GFT and GQT variants can still achieve reasonably strong performance, which also explains why the collapse issue has often been overlooked in prior work.
>
> **Convergence analysis**:
>
> We **ensure all models converge** and **report the best perplexity during training** in Table 1 of the main paper. We additionaly added the plots of reconstruction loss and perplexity during the pretraining process on Cora dataset in **Appendix C (Figure 8)**.
>
> The results confirm that collapse remains an issue for vanilla VQ and several existing anti-collapse techniques. Importantly, collapse does not necessarily lead to large reconstruction errors: a sufficiently strong decoder can overfit node features or links even when only a small subset of tokens is used. This does not mean that low perplexity is acceptable. Rather, when most nodes map to a limited portion of the codebook, the discrete latent space becomes degenerate and fails to capture structural or semantic diversity in the graph. As reflected in Tables 2 and 3, more diverse quantization typically correlates with better downstream performance.
>
> While the optimal codebook size may vary depending on the trade-off between compression and expressiveness, our method provides a flexible and effective framework that maintains token diversity and scales to larger codebooks.

---

> ### Author Response · Authors · 2025-11-27
> **A kind reminder**
>
> Hi reviewer JNRk,
>
> Thank you again for your thoughtful and constructive feedback, which has been very helpful in improving our submission.
>
> As the author–reviewer discussion period is approaching to a close, we would like to kindly check whether you have had a chance to review our detailed rebuttal. We believe that the revisions and explanations we provided directly address the main concerns you raised.
>
> Since these updates and clarifications have made the work more complete and technically comprehensive, we hope they may encourage you to reconsider your evaluation. If you have any remaining questions or concerns, please feel free to let us know at any time, and we will be happy to address them.
>
>
> Best,
>
> Authors

---

> > ### Comment · Reviewer_JNRk · 2025-11-27
> >
> > Thanks for your responses~ I have carefully read your detailed responses (including the common responses quoted from other reviewers) and the corresponding revised paper. Thanks for your extra efforts and the clear clarification of my concerns. Well done and your comprehensive responses really addressed my initial concerns (like the extra validation to support your RGVQ's key story). Currently, from my point of view, I believe this paper is both theoretically and empirically complete, so I'd like to raise my rating and support this paper, leaning towards a clear acceptance.

---

> > > ### Author Response · Authors · 2025-11-27
> > > **Thanks for Your Support**
> > >
> > > Many thanks again for raising score! Your further comments and insights would be invaluable to enhance our work. Thank you once again for your invaluable contribution to our research.
> > >
> > > Best,
> > >
> > > The Authors

---

### Official Review · Reviewer_Bsb5 · 2025-10-28

**Soundness:** 1
**Presentation:** 2
**Contribution:** 2
**Rating:** 2
**Confidence:** 3

**Summary:**

The paper studies codebook collapse in vector-quantized (VQ) tokenizers for graphs and proposes RGVQ, which combines (i) Gumbel-Softmax reparameterization for soft assignments, and (ii) a structure/feature-aware contrastive regularizer that encourages similar nodes to share token distributions and discourages co-assignment for dissimilar nodes.

**Strengths:**

- The motivation is clear. The paper explains that codebook collapse is systemic in Graph VQ and empirically shows large gaps between ideal and observed perplexity across datasets and codebook sizes.
- Results show substantial perplexity gains.
- The paper is well-written.

**Weaknesses:**

- The two building blocks (Gumbel–Softmax and contrastive regularization) are well known; the novelty lies in how they’re used for VQ in graphs and justified by the analysis. This is meaningful but not very novel.
- This paper gives zero attention to mathematical notation. This paper uses indiscriminately all types of letters for all types of elements (sets, vectors, matrices, scalars). Even though one can understand, this hurts the soundness of the paper, and makes the paper not publishable at ICLR.
- The current theoretical discussion is weak. Please (i) instantiate the bounds on real datasets and compare the predicted vs observed codebook perplexity in Fig. 1, and (ii) decompose the bounds’ two terms, quantifying their individual contributions. This would clarify whether the bounds can fully explain the measured perplexities and which factor is the primary driver of imbalance.
- The co-assignment bound is ill-defined and not sound. Under nearest-neighbor quantization, the assignment $z_v$ is a deterministic function of the embedding $h_v$, and $h_v$ itself is deterministic given fixed inputs and parameters; thus $\Pr[z_{v_1}=z_{v_2}]$ is undefined unless a source of randomness is explicitly specified. Moreover, Eq. (27) appears to apply a tail bound (via Markov) and then incorrectly drops the expectation, replacing $\mathbb{E}\,\|h_{v_1}-h_{v_2}\|$ with $\|h_{v_1}-h_{v_2}\|$, which invalidates the inequality. Finally, the geometric implication is stated in the wrong direction: $\|h_{v_1}-h_{v_2}\|\le \delta$ does not guarantee the same codeword in general; the valid statement is that if two nodes share the same token, then their embeddings lie within the cell diameter (e.g., $\|h_{v_1}-h_{v_2}\|\le diam_i$ for codeword $i$).
- Sec 6.2 is not an ablation study but a sensitivity analysis. Is this paper doing model selection with the test set?
-  The “ablation study” suggests that lower Gumbel temperatures yield higher codebook perplexity. This contradicts the paper’s motivation for using Gumbel-Softmax rather than a hard estimator: if lower temperatures help, the STE limit ($\tau \to 0$) should be competitive or even superior. As written, Table 5 indicates that pushing $\tau$ toward zero could further improve perplexity, which contradicts the stated rationale.
- Appendix B shows hyperparameters, but train/validation/test splits, number of seeds, and evaluation protocol for each dataset family are not explicit.

**Questions:**

- Is the observed collapse related to oversmoothing in deeper GNN encoders? Please provide a study of codebook perplexity vs. the number of GNN layers.
- Please clarify precisely how you measure the distance between (L−1)-layer computation trees in (7).
- In (27), why is the expectation dropped? One cannot do that.

---

> ### Author Response · Authors · 2025-11-24
> **Response to the Weaknesses Raised by Reviewer Bsb5**
>
> **Dear Reviewer Bsb5**:
>
> We sincerely appreciate the reviewer’s thoughtful and constructive feedback. Below, we address each weakness carefully.
>
> ### Respond to W1: Novelty and contributions
> We would like to clarify that the main contribution of our work does not lie in introducing building blocks such as Gumbel–Softmax or contrastive learning. Instead, our primary contribution is identifying and explaining a fundamental and previously overlooked problem in Graph VQ.
>
> (1) We reveal that severe codebook collapse is pervasive in real graph datasets and does not improve even when the codebook size increases. Moreover, existing mitigation methods from other domains are largely ineffective. Prior work mainly uses VQ as a module and has not recognized or analyzed this issue, despite its importance for learning both compact and expressive graph tokens.
>
> (2) We provide the first theoretical analysis showing that root cause of collapse is intrinsic to graph data, driven by feature and structure redundancy and self-reinforcing VQ dynamics. This theoretical insight is also validated by our empirical study using easily measurable graph characteristics.
>
> (3) Our solutions are simple but directly motivated by our theoretical analysis, and consistently prevent collapse across all datasets.
>
> ### Respond to W2: Mathematical notations
> We have revised the notation used in this paper for better readability.
>
> ### Respond to W3: Instantiating the bounds on real datasets
> We propose this bound to provide a theoretical foundation for understanding which graph properties are related to the pervasive collapse observed in learning VQ across graph datasets. We do not attempt to numerically separate the two terms in the bound, nor do we use this bound to predict the collapse, since the bound is derived as a conservative condition involving model-dependent Lipschitz constants that cannot be estimated reliably. For the empirical evidence supporting this analysis, please refer to the new experiments discussed in the **General Response Part 3** to this question.
>
> ### Respond to W4 and Q3: Theorem 1
> We really appreciate the viewer for the details. We have carefully revised sections related to Theorem 1 to address the concerns regarding (i) the source of randomness, (ii) the dropped expectation, and (iii) the geometric implication.
>
> (i) Our objective is to define the co-assignment probability that randomly sampled nodes $v_1$ and $v_2$ from the entire graph, i.e., the randomness comes from sampling, not from the encoder and quantizer itself. We have revised the text and formula in Theorem 1 to avoid ambiguity explicitly.
>
> (ii) We acknowledge that the condition requiring node embedding distances to fall within the codeword radius, i.e., $||h_{v_1} - h_{v_2}|| < \delta _c$, does not necessarily guarantee the quantization to the same codeword. However, under a stable quantizer, whenever two sampled embeddings are closer than $\delta _c$, the probability that they are assigned to the same codeword is at least $\alpha$. This assumption is standard in KNN analysis and serves as a mild regularity condition. Accordingly, we have revised Theorem 1 and explicitly introduced the local consistency constant $\alpha$ (Assumption 1 in Appendix B) to reflect this stability. Moreover, boundary cases, where nodes are close but in different Voronoi cells, do exist, but they are difficult to quantify. We exclude them because our goal is to derive a conservative lower bound.
>
> (iii) We acknowledge the oversight of dropping the expectation in Eq. (27) in the original version. We have corrected the derivation and ensured that the expectation term is explicitly preserved in the final bound presented in Theorem 1. Importantly, this correction does not affect the insights and implications provided by Theorem 1.
>
> ###  Respond to W5: Ablation studies and model selection
> For ablation studies, please refer to the details in the **General Response Part 4** to this question. Regarding model selection, we emphasize that our VQ pretraining is fully self-supervised and does not rely on any downstream classification labels. Different checkpoints can be selected based on the unsupervised perplexity measured during the pretraining stage. The pretrained tokens are then integrated into the backbone models (GFT and GQT), which are trained using only the training splits, without accessing any test labels at any stage.
> The downstream experimental settings strictly follow the original protocols of the respective backbone papers.

---

> ### Author Response · Authors · 2025-11-24
> **Response to the Questions Raised by Reviewer Bsb5**
>
> ### Q1 Codebook perplexity vs. the number of GNN layers ($L$).
>
> We provide the quantization results with respect to GNN layers.
>
> | Dataset  | VQ (L=1) | RGVQ (L=1) | VQ (L=2) | RGVQ (L=2) | VQ (L=3) | RGVQ (L=3) | VQ (L=4) | RGVQ (L=4) | VQ (L=5) | RGVQ (L=5) |
> |----------|----------|------------|----------|------------|----------|------------|----------|------------|----------|------------|
> | Cora      | 154.34 | 394.96 | 121.59 | 339.19 | 109.06 | 257.47 | 94.47 | 211.69 | 99.44 | 218.45 |
> | Pubmed    | 8.97   | 452.16 | 3.12   | 300.51 | 5.18   | 295.64 | 4.14  | 319.09 | 4.07  | 295.64 |
> | Photo     | 1.99   | 432.32 | 1.00      | 421.04 | 1.00      | 443.60 | 1.00     | 446.02 | 1.00     | 306.06 |
> | Computer  | 3.81   | 468.98 | 1.00      | 452.41 | 1.00      | 464.65 | 1.00     | 413.10 | 1.00     | 394.83 |
> | Ratings   | 32.64  | 414.10 | 15.59  | 295.28 | 10.80  | 213.42 | 13.29 | 200.93 | 9.14  | 207.18 |
>
>
>
> From this result, we make the following observations: (i) As $L$ increases, the perplexity of vanilla VQ consistently decreases. (ii) RGVQ is robust for different layers because it provides explicit regularization. (iii) These empirical trends are fully aligned with Theorem 1, which relates token co-assignment probabilities to the computation-tree depth $L$.
>
> ### Q2 Distance between computation trees $\Delta^L_{v_1,v_2}$.
> In the original paper, we use the recursive formula in Equation 7:
>
> $p \ge (1 - \frac{1}{\delta_c}[C_1||x_{v_1}-x_{v_2}|| + C_2 \sum_{j\in {\mathcal{N}(v)}}$ $\Delta_{v_1,v_2}^{L-1}])$,
>
> where $\Delta_{v_1,v_2}^{L-1}$ denotes the embedding distance between the $(L-1)$-layer computation trees rooted at the $j$ -th child of \(v_1\) and \(v_2\). It is defined recursively and cannot be computed in a closed form. Instead, in our derivation, we expand $\Delta_{v_1,v_2}^{L-1}$  recursively to the leaf nodes.
>
> For example, in Equation 25, we have
>
> $\Delta^L_{v_1,v_2} \ge C_1||x_{v_1}-x_{v_2}|| + C_2 \sum_{j\in {\mathcal{N}(v)}}$ $\Delta_{v_1,v_2}^{L-1}$ $ \ge 2C_1$ $\mathcal{B}_x  + C_2d_l \Delta _{v_1,v_2}^{L-1}$，
>
> where $\mathcal{B}_x$ is the bound norm of node features.
> By recursively expanding to the leaf nodes, we have:
>
> $\Delta^L_{v_1,v_2} \ge 2\mathcal{B}_x$ $(C_1 + \sum _{l=1}^L C_2^l D_l)$.
>
> Here, $D_l = d_ld_{l-1}...d{1}$ is the total branching factor up to depth $l$, which is related to the degree of nodes.
>
> In summary, we do not explicitly compute the distance between the computation trees; rather, we use the recursive formulation to highlight that this distance depends jointly on the node features and the underlying graph structure. In the revised version of the paper, we have removed this recursive term from Theorem 1 to improve clarity and make the statement easier to understand. We thank the reviewer for this thoughtful feedback.

---

> ### Author Response · Authors · 2025-11-24
>
> ### Respond to W6: Low temperature concern
> We clarify that lower temperatures in the Gumbel–Softmax still yield soft assignment probabilities, so gradients from the regularization flow to all codewords and update unused entries. This mechanism prevents collapse. Although a lower temperature sharpens the distribution, it is fundamentally different from the STE limit, where the assignments become exactly one-hot and inactive codewords receive no gradient updates. Our experiments show that low-temperature Gumbel–Softmax is sufficient to prevent collapse, whereas the hard estimator cannot benefit from this effect.
>
> ### Respond to W7: Hyperparameters and experimental settings
> We provide the dataset-specific hyperparameters of RGVQ in **Appendix B.3 and Table 6** to facilitate reproducibility. Moreover, the evaluation protocols are specified in the captions of the tables.

---

> > ### Comment · Reviewer_Bsb5 · 2025-11-27
> >
> > Thank you for the updates, weaknesses W1, W2, W4, W5, W6, Q1, Q2, and Q3 are now satisfactorily addressed. The overall quality of the paper has increased.
> >
> > I still have the following concerns regarding the revised manuscript and rebuttal:
> >
> > W7: The paper does not specify how the dataset train/validation/test splits were produced, nor how many runs were performed to compute the reported standard deviations. The fine-tuning methodology for node classification remains unclear.
> >
> > Q1: Why is the table or a plot version of it not included in the revised paper?
> >
> > Given these remaining issues, I update my score to 4.

---

> ### Author Response · Authors · 2025-11-27
> **A kind reminder**
>
> Hi reviewer Bsb5,
>
> Thank you again for your thoughtful and constructive comments, which have been helpful for improving our submission.
>
> As the author–reviewer discussion period is coming to an end, we would like to kindly confirm that you have had a chance to review our detailed rebuttal. We believe that the main technical concerns you raised are now thoroughly addressed in our response and revisions.
>
> Additionally, because these updates and clarifications were made specifically in response to your constructive feedback, we kindly hope that they will encourage you to reconsider your evaluation. Should you have any remaining questions or reservations, please feel free to let us know at any time. We are fully committed to addressing all concerns to your satisfaction.
>
> Best,
>
> Authors

---

> ### Author Response · Authors · 2025-11-28
> **Response to Reviewer Bsb5: W7 and Q1 (Part1/3)**
>
> Dear reviewer Bsb5,
>
> We sincerely appreciate your further feedback. We first address the experimental settings and hyperparameters (W7) and then answer Q1.
>
> ### W7: Implementation details and hyperparameters
>
>
> ### Codebook utilization
>
> For RGVQ, we set the codebook size $K$ to 512. To construct the contrastive sample sets, for each node, we construct a pool of positive candidates by combining its 1-hop neighbors with the nodes that are most 20 similar in the input feature space (top-$K=20$), which is measured by cosine similarity. From this pool, we sample 50 nodes with replacement as positive samples. The negative pool is defined symmetrically as all nodes that are neither neighbors nor feature positives, and we sample 50 negative nodes with replacement from the negative pool. The training weights for the link reconstruction, node feature reconstruction, contrastive regularization, commitment loss, and vocabulary loss are set to 0.01, 100, 1, 0.1, and 0.9, respectively. We set the temperature for the Gumbel-Softmax trick to 0.1.
> **We train on each dataset for 1000 epochs to ensure convergence, and repeat the process 20 times to report the mean perplexity with standard deviations.** Detailed hyperparameters for each dataset are summarized in **Table 6 in Appendix B**.
>
>
> For finetuning tasks, we strictly follow the original settings in the paper of GQT[1] and GFT[2]. We have added the details of the implementation, runs, and dataset splits of the experiments for downstream tasks in **Appendix B.3** in the manuscript, and we also detail them below:
>
> ### Transferability
>
> We use RGVQ as a plugin within the pretraining pipeline of GFT.
>
> Specifically, we retain the same pretraining tasks in GFT [2], including the link, node feature, and node embedding reconstruction tasks, and integrate RGVQ as a regularization term. Their weights are set to 100, 1, 0.01, and 10, respectively. For the backbone encoder, we utilize a 2-layer GCN model with ReLU activation, and set the codebook size to 512 and the hidden dimension to 256. We use AdamW optimizer with a learning rate of 1e-3 and weight decay of 1e-5. For data augmentation, we apply a link drop rate and the node-feature drop rate of 0.2. We pretrain the VQ tokens for 500 epochs on all datasets. During finetuning, **we repeat each experiment 20 times** to report the average performance with standard deviations. We finetune the model for 250 epochs using early stopping.
>
> For dataset splits, we follow the commonly used protocol for Cora and PubMed and utilize the predefined 10 splits with different seeds to report the downstream performance. Each split includes 20 labeled nodes per class for training. For WikiCS, we follow the recommended protocol by OGB and use the official split, reporting average performance across 20 splits [2]. For WN18RR, we utilize 86,835/3,034/3,134 links for training/validation/test, respectively. For FB15K237, we use 272,155/17,535/20,466 links for training/validation/test, respectively. For HIV and PCBA, we follow the official data split and utilize 80\%/10\%/10\% for training/validation/test set [3].

---

> ### Author Response · Authors · 2025-11-28
> **Response to Reviewer Bsb5: W7 and Q1 (Part2/3)**
>
> ### Serialization
>
> The training of GQT includes two parts: the VQ tokenizer and the backbone transformer.
>
> For the VQ tokenizer, we follow the original paper and use Residual VQ [1]. We retain all of the reconstruction tasks in the pretraining setting of GQT, including Deep Graph Infomax (DGI) and GraphMAE2, and integrate RGVQ as a regularization term. For the tokenizer, we set the number of codebooks to three for GQT, GQT + EMA, GQT + AP, GQT + Reset, GQT + PT; and one for GQT + SimVQ, GQT + RGVQ. We choose codebook size from \{128,256,512\}. For the GNN encoder, we adopt GCN with ReLU activation, varying the number of layers from \{2,3,4,6\} and hidden dimensions from \{256,512\}. We pretrain the VQ tokenizer and the GNN encoder for 200 epochs until convergence.
>
> For training the vanilla transformer, we construct semantic links using K-Nearest-Neighbors, with K in \{0,5,10\}. To serialize the input graph sequence, we use Personalized PageRank (PPR) to generate a sequence for each node, with the sequence length selected from \{15,20,30\}
> The transformer uses 2 or 3 layers, 4 attention heads, and a feedforward dimension of \{512, 1024\}.  We train transformers with node labels together with the pretrained VQ tokenizer and GNN encoder, and **report the average performance and standard deviations over 5 runs**.
>
> For Cora, Pubmed, Citeseer, Computer, Photo, we follow the original settings in GQT, using 60\%/20\%/20\% for training/validation/test. For WikiCS, we follow the predefined split in the original paper [4] and report the average performance across 20 splits. For Amazon-Ratings, Roman-Empire, and Questions, we adopt the splits in the original paper[5], using 50\%/25\%/25\% for training/validation/test, and report the mean performance over 10 random splits. The detailed hyperparameters are summarized in the Table below.
>
> | Dataset | GNN: #Layers | GNN: Hidden Dim | Quantizer: #Codebooks | Quantizer: Codebook Size | Transformer: KNN | Transformer: PPR | Transformer: #Layers | Transformer: #Heads | Transformer: FFN Dim |
> | --- | --- | --- | --- | --- | --- | --- | --- | --- | --- |
> | Cora | 2 | 256 | 3 | 128 | 0 | 15 | 2 | 4 | 512 |
> | CiteSeer | 2 | 256 | 3 | 128 | 5 | 15 | 2 | 4 | 512 |
> | PubMed | 2 | 256 | 3 | 256 | 0 | 15 | 2 | 4 | 512 |
> | Computer | 2 | 256 | 3 | 128 | 5 | 30 | 2 | 4 | 512 |
> | Photo | 3 | 512 | 3 | 128 | 5 | 20 | 2 | 4 | 1024 |
> | WikiCS | 2 | 256 | 3 | 128 | 5 | 30 | 2 | 4 | 512 |
> | Amazon-Ratings | 4 | 512 | 3 | 128 | 5 | 20 | 2 | 4 | 1024 |
> | Roman-Empire | 6 | 256 | 3 | 256 | 10 | 15 | 3 | 4 | 512 |
> | Questions | 3 | 256 | 3 | 512 | 10 | 15 | 2 | 4 | 512 |
>
> ### Q1: Table of GNN layers is not included in the revised paper.
>
> We apologize for our oversight in not including this additional experiment. We have now revised the paper, and the results with analysis are provided in **Influence of GNN Layer Number and Table 8** in **Appendix C**.

---

> ### Author Response · Authors · 2025-11-28
> **Response to Reviewer Bsb5: W7 and Q1 (Part3/3)**
>
> We believe the additional experiment makes our paper more empirically complete, and the detailed experimental setup has ensured the reproducibility of our results.
>
> We hope that our response addresses your concerns, and we look forward to your reply.
>
> Reference
> [1] Learning graph quantized tokenizers. ICLR’25
>
> [2] Gft: Graph foundation model with transferable tree vocabulary. NeurIPS’24
>
> [3] Open graph benchmark: Datasets for machine learning on graphs. NeurIPS’20
>
> [4] Wiki-cs: A wikipedia-based benchmark for graph neural networks. ICML’20
>
> [5] A critical look at the evaluation of GNNs under heterophily: Are we really making progress? ICLR’23

---

### Official Review · Reviewer_f6SV · 2025-11-02

**Soundness:** 3
**Presentation:** 2
**Contribution:** 3
**Rating:** 6
**Confidence:** 3

**Summary:**

The article is a work aimed at contributing to the design of graph foundation model, by studying carefully how vector quantisation can be used to learn graph tokens. For that, the authors study limitations of previous use of vector quantisation (VQ) for graph representations, showing that their main limits is codebook collapse and that it’s caused by 2 things: imbalances in early assignments to codewords (due to similarities in attributes or structures), and  the training dynamics which removes non-used (and even less-used) codewords and does not allow them to be used again. The authors provide both theoretical insights and experimental evidences on that.

To alleviate these difficulties, the authors design a new approach of VQ training which includes a regularisation over the graph, so that the codebook collapse doesn’t appear. The method is relevant and elegant. A section with numerical experiments study the obtained gains and performance, and its is shown that, on classical examples, the proposed RGVQ representation works better than previous solutions to avoid codebook collapse, and that it has transferability properties (hence paving the way to graph foundation models).

**Strengths:**

The strengths of the article are :

1. A very good analysis of the limitations of VG for graphs as used before, with numerical experiments reporting the codebook perplexity and which is far from optimal in existing methods.

2. This analysis comes also with a nice theoretical insight (with Theorem 1) to explain that, in addition to insights about the self-reinforcing training which removes less-used codewords from the training and enforces this collapse by preventing unused codeword to be considered again.

3. The proposed method, called RGVQ (for regularised graph) is not overly complex yet it does the work of preventing the codebook collapse.

4. The numerical experiments show that the method works well, better than previous propositions for a better VQ for graphs.

All in all, I found that the contributions are correct, and that there is enough novelty and insight to be presented at this conference. However, it comes with some weakness, especially in the presentation, as stated underneath.

**Weaknesses:**

The main weakness of the article are about the presentation, and here are the points which should be improved according to me:

1. For me, section 3 about the VQ construction was hard to follow. Specifically, the notation $z_i = \delta_j \mathbf{C}$ used for eq. (3) had me go back to referenced work to understand how the sg operator comes in to eq. (3).  I think that presenting first the loss of eq. (4) with the sg operator and explaining then how sg works and why we need these three parts in the loss, should be clearer.

2. Also, the presence of D and \sigma in eq. (5) is not really commented. Are specific forms assumed ? The authors refer to articles about other forms of losses. It would be good to known whether the present work could use various forms of D, other losses, and possibly other ways to encode A (why use \sigma(z z^T) necessarily?).

3. I am not certain of the use of the notation in eq. (10) : $\pi  =  - \Vert h_i - \mathbf{C} \Vert^2$. I would have expected that $\pi$ is here some p_i(e_j \ h_i) and I would not have written it the way it is in eq. (10)

4. The proposed RGVQ is compared against other methods with VQ with mitigation of the difficulties. But the baselines include no other attempts at graph foundation models, which currently achieve baseline close (or slightly better) than the proposed RGVQ with GFT. Two examples could be (chosen because I have seen their results and because they have some of their dataset in common with the present work): ULTRA by Galkin et al. ICLR 2024, or SCORE, by Wang and Luo, 2024.

This is not to say that the present contribution is not valuable ; this is more to show that existing (and different) approaches to graph foundation models provide almost similar performance, at least on node classification.

5. The graphs in Figure 4 are too small to be read on papers; please enlarge the legends (and use symbols).

6. Section 6.4 does not strike me as an ablation study, more as a study on the impact of the hyperparameters. Also I have questions: when n increases, the perplexity decreases, why ? Why also is there a dependency on the dataset in 4 (b) ? For figure 4 (c) : why not report also perplexity for this numerical experiment ? For 4(d) about diversity: what procedure is used to vary the obtained perplexity ?


References:


Kai Wang and Siqiang Luo. Towards graph foundation models: The perspective of zero-shot reasoning on knowledge graphs. arXiv preprint arXiv:2410.12609, 2024.


M Galkin, X Yuan, H Mostafa, J Tang, and Z Zhu. Towards foundation models for knowledge graph reasoning. International Conference on Learning Representations, 2024.

**Questions:**

Some additional questions:

* You could remind the bounds of perplexity and its link to entropy in 3. Also, one often uses distorsion to quantify the use of a quantizer and of codebook; would it make sense here ?

* I have carefully checked the proof of theorem 1 and it seems ok for me. The bound in eq. (23) could be commented: is it expected to be tight or realistic on average ? or often an overestimation ?  This would help to know whether the bound of the theorem (or eq. (28))) is really a conservative estimate or not.

* About theorem 1: it could be interesting to comment about the two limits when B_x goes to zero and when \Delta_l becomes large.

* In section 5, provide at least one reference for InfoNCE, and possibly some rationale for using it.

---

> ### Author Response · Authors · 2025-11-24
> **Response to the Weaknesses Raised by Reviewer f6SV**
>
> We sincerely thank the reviewer for the time and effort in reviewing our paper, and for the valuable feedback. Please see below for our responses to your comments.
>
> ### Response to W1, W2, W3, and W5: Notations and Figures
> We thank the reviewer for this suggestion. We have revised the related text to make it easy to understand, and would like to clarify the function of $\text{sg}[\cdot]$ and the loss function of VQ here.
>
> On the stop-gradient sg[$\cdot$] :
> In deterministic VQ, we apply a nearest codebook selection $k = \arg\min_j ||h_i - e_j||^2,   z_i = e_k$, which is non-differentiable, so $∂z_i / ∂h_i = 0.$ To ensure the gradient can be propagated to the encoder, we apply the Straight-Through Estimator (STE) to estimate the gradient:
>
> $z_i = \text{sg}(e_k - h_i) + h_i,
> $
>
> which blocks gradients through the discrete codeword selection (which is non-differentiable) while still letting gradients pass to the encoder, yielding $∂z_i / ∂h_i = 1.$
>
> On the three losses:
> The reconstruction loss trains the encoder and decoder to compress and reconstruct the input samples, the codebook loss pulls selected codewords toward encoder outputs, and the commitment loss pulls encoder outputs toward the quantized representations to keep them aligned.
>
> On the forms of link and feature reconstruction:
> In our work, we do not assume any specific form of loss functions for either link or feature reconstruction. We did extensive experiments with multiple loss formulations and found that the collapse phenomenon is not caused by the choice of loss function forms, but rather by the intrinsic properties of graph data, i.e., feature and structural redundancy. To avoid potential ambiguity, we have revised the text accordingly and updated Equation (5).
>
> On the notation of $\pi$:
> We use $\pi _i= -||h_i - \mathbf{C} ||$ as a simplified notation to represent the distance between the node embedding $h_i$ and all codewords in the codebook $\mathbf{C}$. Based on this distance vector, we apply a softmax to obtain the token assignment distribution $p_i(\mathbf{C} \mid h_i) = \text{Softmax}(\pi _i)$. Here, $\pi _i$ is introduced purely for notational convenience.
>
> On the size of Figure 4:
> We sincerely appreciate the reviewer's suggestion. We have made it clearer and moved the subgraph Figure 4 (d) in the original version to the appendix.
>
> ### Response to W4:
> We thank the reviewer for the concern regarding the marginal improvements in node classification compared with recent graph foundation models. We would like to restate our motivation: First, the primary purpose of Graph VQ is to obtain discrete graph tokens that (i) serve as transferable vocabularies for graph foundation models, (ii) compress graph information, and (iii) enable sequence modeling in transformers. In this context, our goal is to address an ignored but fundamental issue in graph VQ, i.e., codebook collapse, which severely limits the usability of VQ for graph tokenization.
>
> Second, our downstream results are obtained using a VQ-based GFM and Transformer backbone, and thus the absolute performance depends on these backbones rather than the proposed RGVQ module alone. The core contribution of RGVQ is improving VQ stability and token quality, not a new GFM architecture.
>
> ### Response to W6: ablation studies
> We acknowledge that our ablation analysis is conducted through hyperparameter studies. Since the two proposed modules, i.e., Gumbel-Softmax reparameterization and structure-aware regularization, are mutually independent, removing either one would cause RGVQ to degenerate into vanilla VQ. Therefore, we vary the key hyperparameters in each module to evaluate their individual contributions. In addition, we have added a new drop-one ablation to further address the reviewers’ concerns. Please refer to **General Response Part 4** for more details.
>
> Q: When the sample number increases, why does the perplexity decrease?
> A: A larger sample number introduces noisier or less relevant contrastive pairs, weakening the regularization signal and reducing effective codebook usage, which leads to lower perplexity.
>
> Q: Why is there also a dependency on the dataset in 4 (b) and (c) :
> A: Different datasets have different levels of structural and feature redundancy; thus, we varied the critical hyperparameters in three datasets to assess the performance of RGVQ.
>
> Q: Why not report also perplexity for this numerical experiment?
> A: We report the line figures to illustrate the trend of quantization results with respect to the key hyperparameters more clearly.
>
> Q: For 4(d) about diversity: what procedure is used to vary the obtained perplexity?
> A: During pretraining, perplexity gradually increases and stabilizes at an optimal level, as shown in Figure 8 in Appendix C. We select checkpoints with varying perplexities and utilize them for downstream tasks to report their corresponding performance.

---

> ### Author Response · Authors · 2025-11-24
> **Response to the Questions Raised by Reviewer f6SV**
>
> ### Response to Q1: using distortion as the indicator
>
> We thank the reviewer for the suggestion. Distortion is a standard measure in classical vector quantization. However, in graph VQ, the objective is not reconstruction but discrete representation learning, where the codebook is used for downstream tasks rather than merely minimizing reconstruction loss. Therefore, distortion is not directly aligned with our goal. Instead, perplexity more meaningfully reflects collapse and token utilization in our setting. Additionally, we added a new experiment, which reports the reconstruction loss of different baselines during training in Figure 8 (Appendix C). Please refer to the added experiments for more details.
>
> ### Response to Q2 and Q3: Theorem 1.
> We thank the reviewer for constructive suggestions for commenting on Theorem 1.
> Regarding the bound of distance between node embeddings:
>
> $
> \Delta^L_{v_1,v_2} \ge 2\mathcal{B}_x(C_1 + \sum _{l=1}^L C_2^l D_l).
> $
>
> It is a very loose overestimation. This is because: When deriving the bound, we repeatedly used the maximum feature norm $\mathcal{B}_x$ and the maximum branching number of each layer of the computation tree $l$. The actual feature norm and branch number should be smaller. Thus, this recursive use of maxima makes the distance bound an overestimation.
>
> More importantly, Theorem 1 is the most conservative lower bound of token co-assignment probability. The actual probability is always higher than this closed form. Therefore, due to the loose nature,  this theory is not supposed to predict how severe the collapse will be (e.g., when $\mathcal B_x \to 0$ or when $\mathbb E(D_l)$ becomes large). Instead, it shows that the common collapse in Graph VQ is influenced by both feature redundancy and structure redundancy. Please refer to Appendix A for more details.
>
> ### Response to Q4: Reference of InfoNCE
> We thank the reviewer for this suggestion. We have added two references for InfoNCE.

---

### Author Response · Authors · 2025-11-23
**General Response to AC and Reviewers - Part 1: Manuscript Revision**

We sincerely thank all the reviewers for their constructive feedback and insightful comments. We are pleased that the reviewers recognized the significance and contribution of our work. We acknowledge the common concerns raised by the reviewers and would like to address them in this general response. Additionally, we have updated the paper based on the feedback, and all changes are highlighted in **red** in the updated manuscript.



The changes are summarized as follows:

1. We modified Theorem 1 and its derivation to improve correctness and clarity (Reviewer Bsb5).
1. We modified the mathematical notations for better readability (Reviewer f6SV and Bsb5)
2. We include the pesoducode for the training procedure for RGVQ to address the ambiguity in the actual training loss (Reviewer JNRk);
3. We add more baselines in the experiments, including GFT with anti-collapse methods (Table 2) and GQT with anti-collapse methods (Table 3), and reconstruction loss and perplexity during pretraining (Figure 8) (Reviewer JNRk).
4. We change the location of the table of the Drop-one ablation study (Table 4) and Correlation between downstream performance and codebook perplexity(Figure 9) for better readability and more complete ablation studies (Reviewer f6SV, Bsb5, and 8u7x)
5. We add the experiment of influence of GNN layers (Appendix C and Table 8) and detailed experimental settings regarding finetuning tasks (Appendix B.3) (Reviewer Bsb5).
6. We add the experiments of instantiating Theorem 1 on real datasets (Appendix C and Table 9).

---

> ### Author Response · Authors · 2025-11-23
> **Part 2: Theorem 1 and its derivation**
>
> We have carefully revised sections related to Theorem 1 to address the concerns regarding (i) the source of randomness, (ii) the dropped expectation, and (iii) the geometric implication.
>
> (i) Our objective is to define the co-assignment probability that randomly sampled nodes $v_1$ and $v_2$ from the entire graph, i.e., the randomness comes from sampling, not from the encoder and quantizer itself. We have revised the text and formula in Theorem 1 to avoid ambiguity explicitly.
>
> (ii) We thank the reviewer Bsb5 for pointing out that the condition requiring node embedding distances to fall within the codeword radius, i.e., $||h_{v_1} - h_{v_2}|| < \delta_c$, does not necessarily guarantee the quantization to the same codeword. However, under a stable quantizer, whenever two sampled embeddings are closer than $\delta_c$, the probability that they are assigned to the same codeword is at least $\alpha$. This assumption is standard in KNN analysis and serves as a mild regularity condition. Accordingly, we have revised Theorem 1 and explicitly introduced the local consistency constant $\alpha$ (Assumption 1 in Appendix B) to reflect this stability. Moreover, boundary cases, where nodes are close but in different Voronoi cells, do exist, but they are difficult to quantify. We exclude them because our goal is to derive a conservative lower bound.
>
> (iii) We thank the reviewer Bsb5 for pointing out the oversight of dropping the expectation in Eq. (27) in the original version. We have corrected the derivation and ensured that the expectation term is explicitly preserved in the final bound presented in Theorem 1. Importantly, this correction does not affect the insights and implications provided by Theorem 1.

---

> ### Author Response · Authors · 2025-11-23
> **Part 3: Instantiating the bounds on real datasets (Reviewer Bsb5 and JNRk)**
>
> We propose this bound to identify which graph properties could lead to collapse. It reveals the co-assignment probability is related to two data-dependent components: a feature term involving the feature norm $\mathcal{B}_x$, and a structure term involving the computation-tree expansion $\mathbb{E}[D_l]$. Other constants, such as $C_1$ and $C_2^l$, are model-dependent Lipschitz constants that cannot be estimated reliably. Thus, we use PCA@95 as a proxy for effective feature variation and the average node degree as a proxy for the expectation of the degrees of computation trees.
>
> **Table: Dataset statistics (PCA@95, average degree) and measured codebook perplexity across 8 graph datasets.**
>
> | **Dataset** | **Cora** | **Pubmed** | **Citeseer** | **Photo** | **Computer** | **Ratings** | **Roman** | **Questions** |
> |-------------|----------|------------|--------------|-----------|--------------|-------------|-----------|----------------|
> | **PCA@95**      | 802 | 410 | 1459 | 611 | 646 | 194 | 141 | 160 |
> | **Avg Degree**  | 4.9 | 5.5 | 3.74 | 32.13 | 36.76 | 8.6 | 3.91 | 7.28 |
> | **Perplexity**  | 94.47 | 4.14 | 60.09 | 1.00 | 1.00 | 13.29 | 10.84 | 20.78 |
>
>
> Across all 8 datasets, these characteristics align with the quantization results of vanilla VQ: datasets with high average degree (Photo, Computer) or low PCA@95 (Ratings, Roman, Questions) exhibit lower perplexity, whereas datasets with both higher PCA@95 and lower degree (Cora, Citeseer) exhibit much weaker collapse. This supports our theoretical insight that feature redundancy and structural redundancy drive collapse in Graph VQ.
>
> We do not attempt to numerically separate the two terms in the bound, nor do we use it to predict the collapse.

---

> ### Author Response · Authors · 2025-11-23
> **Part 4: Ablation study (Reviewer f6SV, Bsb5, and 8u7x)**
>
> We clarify that we use the sensitivity analysis in Sec 6.2 as ablation studies because the two proposed components are not independent, and removing either would make the other ineffective. The structure-aware regularization depends on the soft assignment probabilities produced by the Gumbel–Softmax reparameterization; without reparameterization, one-hot assignments provide no gradient to inactive codewords, rendering the regularization ineffective. Conversely, using soft assignments without regularization behaves like vanilla VQ and fails to prevent collapse. Therefore, we choose to vary the important parameters to control the impacts of each proposed module to assess their contributions. For example, we vary the temperature (controlling the sharpness and gradient flow of soft assignments) and the number of contrastive samples (controlling the strength of the regularization).
>
> However, we include a Drop-one ablation study which assesses three variants: removing all proposed components (variant-1), reparameterization with only structurally similar samples (variant-2), and reparameterization with only feature-similar samples (variant-3).
>
> | Variant   | Gumbel-Softmax | Structure samples | Feature samples | Cora Perp. | Cora Acc. | PubMed Perp. | PubMed Acc. | WikiCS Perp. | WikiCS Acc. |
> |-----------|----------------|-------------------|------------------|------------|-----------|---------------|--------------|---------------|--------------|
> | Variant-1 |                |                   |                  | 94.47      | 78.35     | 4.14          | 73.39        | 10.18         | 79.13        |
> | Variant-2 | ✓              | ✓                 |                  | 172.32     | 79.87     | 215.35        | 76.32        | 153.35        | 79.84        |
> | Variant-3 | ✓              |                   | ✓                | 135.45     | 79.12     | 208.16        | 76.29        | 179.49        | 79.79        |
> | RGVQ      | ✓              | ✓                 | ✓                | **211.69** | **80.85** | **319.09**    | **77.46**    | **228.82**    | **80.10**    |
>
> It is obvous that excluding either the structural-similar or feature-similar set reduces quantization diversity and consequently harms downstream accuracy.
> Removing Gumbel-softmax causes RGVQ to degenerate to vanilla VQ and leads to severe collapse.
> These results indicate that Gumbel-Softmax reparameterization and structure-aware regularization are mutually dependent in preventing codebook collapse, and both topological and feature information are essential for enhancing quantization diversity.
>
> This table has been included in the main text in Section 6.2 in the revised paper.

---

> ### Author Response · Authors · 2025-11-23
> **Part 5: Scalability and computation overhead (Reviewer JNRk and 8u7x)**
>
> We thank the reviewers for raising concerns regarding the scalability to large graphs and computation overhead.
>
> In Appendix D, we provide the complexity analysis of RGVQ, which includes pre-computation of the contrastive set and quantization process. In practice, for the pre-computation of the contrastive set, we adopt a sampling strategy: for each node, we sample $M$ non-neighbor nodes (where $M$ is a small constant, e.g., 100) and compute their feature similarity. This limits the total cost of semantic similarity computation to $O(|\mathcal{V}| \cdot M \cdot d)$, which is linear in the number of nodes $|\mathcal{V}|$ and feature dimension $d$.
> After collecting both structurally and semantically similar candidates, we perform top-$k$ selection for each node to finalize its positive sample set, costing $O(|\mathcal{V}| \log k)$ time in total. The overall time complexity of the contrastive set construction process is  $O(|\mathcal{E}| + |\mathcal{V}| \cdot M \cdot d + |\mathcal{V}| \log k)$, which is linear in the number of nodes $|\mathcal{V}|$ and edges $|\mathcal{E}|$ under fixed $M$ and $k$.
> Moreover, RGVQ uses the InfoNCE loss between node pairs to regularize the token assignment distributions. For each node, this involves computing similarities with $k$ positive and $k$ negative samples, each over $K$-dimensional distributions. The total cost is $O(|\mathcal{V}| \cdot (2k) \cdot K)$.
>
> Our model is easy to scale to large graphs, as we adopt the same strategy for each minibatch. For example, we implement the same sampling strategy and construct the contrastive set in each input mini-batch. We additionally include the performance of RGVQ against vanilla VQ on ogbn-arxiv, ogbn-product, and ogbn-proteins
>
> | Method | ogbn-arxiv        | ogbn-products      | ogbn-proteins       |
> |--------|--------------------|---------------------|----------------------|
> | VQ     | 6.43 ± 0.94       | 10.13 ± 1.05       | 9.85 ± 0.86          |
> | RGVQ   | 305.13 ± 20.3     | 223.47 ± 10.81     | 254.87 ± 0.67        |

---

### Author Response · Authors · 2025-11-29
**Rebuttal Summary for Area Chair**

Dear Area Chair,

Thank you so much for reviewing our paper and the discussion process of our submission.

Here, we would like to briefly summarize the key progress we made during the discussion period.

Our paper initially received four reviews with scores 6 (f6SV), 2 (Bsb5), 4 (JNRk), and 6 (8u7x). During the discussion period, we carefully addressed every concern and question raised by all reviewers and updated the manuscript accordingly. Three of the reviewers provided feedback on our rebuttal and updated their scores **before the system bug occurred**, as follows:

- Reviewer **JNRk** increased their score 4 → **8**, shifting from borderline reject to a clear accept (26 Nov, 2025 18:52  AOE Time).
- Reviewer **Bsb5** increased their score 2 → **4**, shifting from reject to borderline (26 Nov, 2025 20:28  AOE Time).
- Reviewers **8u7x** maintained **6** as their original positive scores (24 Nov, 2025 18:49  AOE Time).
- Reviewers **f6SV** gave us a original positive score **6** and have not responded to our rebuttal.

Reviewer **Bsb5**, who increased their score from 2 to 4, also raised two additional questions about the parameter settings at that time (26 Nov, 2025 20:28  AOE Time). We promptly provided detailed clarifications and updated the manuscript accordingly, but the discussion ended before we could receive their follow-up.

---

> ### Author Response · Authors · 2025-11-30
> **Summary of Responses for Each Reviewer (Part 1/2)**
>
> To accelerate and assist your review process, we provide below a concise summary of the reviewers’ main concerns and how we addressed each of them. Moreover, commonly raised concerns by the four reviewers are addressed in **Global Response**.
>
> ### **Reviewer JNRk (Score: 4 → 8)**
>
> - **Theory-to-method Gap (W1 and Q4).**
>
>     We instantiated Theorem 1 on real datasets and connected node features and graph connectivity to the observed codebook collapse. We also provided the detailed gradient analysis of two proposed component to prove that our method fundamentally breaks the dynamics of conventional VQ to prevent collapse.
>
> - **Ambiguity in the Actual Training Loss (W2 and Q1).**
>
>     We added a pseudocode of the training process in the main text, and revised the actual training loss (Equation 16) to improve clarity.
>
> - **Structure-aware Contrastive Term Underspecified and Scalability (W3, W5, and Q5).**
>
>     We provided detailed complexity analysis, detailed descriptions of the sampling procedure, and additional results on large graphs to the reviewer and General Response Part 5, showing the effectiveness and scalability of our model.
>
> - **Baseline Fairness Concerns (W4, Q2, and Q3).**
>
>     We added extensive baselines in the experiments, including GFT and GQT with anti-collapse variants (Tables 2 and 3), showing that our method not only ensures better codebook utilization but also yields improvements on downstream tasks. Moreover, we clarified that all models are fully converged and reported using best perplexity for fairness. We also provided the plots of reconstruction loss and perplexity during the pretraining process in Appendix C (Figure 8), confirming that collapse persists in vanilla VQ and several existing anti-collapse baselines.
>
>
> The reviewer responded: **"Well done and your comprehensive responses really addressed my initial concerns. I believe this paper is both theoretically and empirically complete, so I'd like to raise my rating and support this paper, leaning towards a clear acceptance."**
>
>
> ### **Reviewer Bsb5 (Score: 2 → 4)**
>
> -  **Limited Novelty (W1)**
>
>     We clarified that our contribution is not simply combining two components. Instead, through extensive experiments, we identified an overlooked issue in graph VQ: the pervasive collapse and the ineffectiveness of existing mitigation strategies. We further showed that the root causes lie in graph-specific properties and VQ dynamics via theoretical analysis and empirical evidence. These insights motivated our proposed regularization term, which is simple yet effective.
>
> - **Notation Clarity (W2):**
>
>   We revised the notation in this paper for better readability.
>
> - **Discussion and Soundness of Theorem 1 (W3, W4, and Q3)**:
>
>    We realized that the reviewer’s concerns mainly came from ambiguities in where the randomness comes from and how the geometric implication should be interpreted. To address this, we clarified the sampling randomness, restored the missing expectation, introduced a clear local-consistency assumption, and fixed the errors the reviewer pointed out. We also added an instantiation of Theorem 1 on real datasets in the global response to show how the theory connects to practice. These clarifications do not change the motivation or the insights of Theorem 1.
>
> -  **Ablation Study (W5, W6)**
>
>    We explained why hyperparameter analyses were initially used and clarified the experimental settings related to model selection, as well as the fundamental differences between low-temperature behavior and deterministic VQ. We also added a drop-one ablation in the revised version to fully address the reviewer’s concerns.
>
> -  **Explanation of Computation-Tree Distances (Q2)**
>
>    We clarified the motivation for introducing the $L-1$ layer computation trees and explained how they are recursively expanded into the final form of Theorem 1. The corresponding text and theorem were revised for clarity.
>
> The reviewer claimed: **“Weaknesses W1, W2, W4, W5, W6, Q1, Q2, and Q3 are now satisfactorily addressed. The overall quality of the paper has increased. Given issues of Q1 and W7, I update my score to 4.”** We accordingly responded and added these details to our paper and have not got the feedback from the reviewer.
>
> -  **Collapse and Over-smoothing (Q1):**
>
>    We added new experiments showing how perplexity varies with the number of GNN layers, demonstrating that deeper GNNs are more prone to collapse. This has been included in the revised manuscript.
>
> -  **Experimental Setting and Hyperparameters (W7)**
>
>    We added detailed experimental settings for both quantization and fine-tuning, including hyperparameter tables and dataset train/test splits, in the response and Appendix B.

---

> ### Author Response · Authors · 2025-11-30
> **Summary of Responses for Each Reviewer (Part 2/2)**
>
> ### **Reviewer 8u7x (Score: 6 → 6)**
>
> **Concerns:**
>
> - **Scalability (W1, W3, and Q1)**
>
>     We provided detailed complexity analysis, detailed descriptions of the sampling procedure, and additional results on large graphs to the reviewer and General Response Part 5, showing the effectiveness and scalability of our model.
>
> - **Ablation Depth (W3 and Q3)**
>
>     We added additional ablations, including temperature study showing a consistent correlation between higher perplexity and better downstream accuracy, as well as a drop-one ablation. Based on the complexity analysis, we also clarified that temperature does not influence the computation overhead, while the complexity of the construction of contrastive sets and InfoNCE regularization are both linear to the number of contrastive samples.
>
> - **Generality of the “natural regularization” claim (W4)**
>
>     We clarified the connection between our empirical observations, theoretical analysis, and the proposed components, showing that the method explicitly regularizes token assignment distributions through feature similarity and graph connectivity. In this sense, we refer to “graph as a natural regularization.
>
> - **Hyperlinks (W5)**
>
>     We checked all the hyperlinks and made sure they work in the revised version.
>
> - **Extension to other graphs (Q2)**
>
>     We explained that our framework can be easily extended to different graphs because both key components are GNN-agnostic and only require node embeddings and the computation of feature similarity.
>
> - **More Baselines, integrating RGVQ in othe GFMs (W2 and Q4)**
>
>     We clarified that our method is designed to specifically address the token learning problem in Graph VQ models and extending RGVQ to such architectures would require adapting or introducing VQ-based token modules
>
>
> The reviewer replied: "**The authors provide a positive and detailed rebuttal, I therefore maintain my original score.**"
>
>
> ### **Reviewer f6SV (Score: 6 → 6)**
>
> - **Notation and equation clarity (W1, W2, W3, W5):**
>
>     We clarified all notations to the reviewer and revised the corresponding equations and figures to improve readability.
>
> - **Missing More Graph Foundation Model baselines (W4):**
>
>     We explained that our contribution aims to improve VQ quantization and performance of VQ-based models, not proposing a new GFM. Therefore, adding non-VQ GFMs as baselines is misaligned with our motivation.
>
> - **Ablation study (W6):**
>
>     We explained why hyperparameter analyses were initially provided as the ablation. We have now added a drop-one ablation in the revised version and addressed the reviewer’s three questions regarding the ablation results.
>
> - **Alternative quantization metrics (Q1):**
>
>     We clarified that Graph VQ focuses on discrete representations for downstream tasks, not reconstruction. Thus the suggested metrics do not align with our objective, but we added an additional reconstruction experiment in the Appendix C for completeness.
>
> - **Comments on Theorem 1 (Q2, Q3):**
>
>     We clarified that the bound is intentionally loose and conservative, aiming to illustrate the relationship between graph redundancy and codebook collapse rather than to predict precise collapse trends.
>
>
> Finally, we would like to thank the AC again for your time and efforts in reviewing our submission.
>
> The Authors

---

### Meta-Review · Area_Chair_pzeJ · 2025-12-17

**Summary:**

After reading the paper, reviews, and rebuttals carefully, I acknowledge that many concerns of the reviewers have been addressed by the rebuttal and revision. However, there are still several significant limitations, most of which were not discovered by the reviewers.
* The correctness of the theoretical results (as pointed out by Reviewer Bsb5) is questionable. The authors' rebuttals seem failed to address this. In addition to the comments of Reviewer Bsb5, I must point out the following issues regarding Theorem 1:
    * $W_1$ and $W_2$ are not comparable to the depth of the GNN, which has $L$ layers.
    * "Lipschitz term" isn't a formal terminology.
    * $\delta_c$ hasn't been defined in or around the theorem.
    * The bound becomes trivial ($p$ is greater than a negative value) when $L, C_1, C_2$ are not small enough and $\delta_c$ is not large enough.
    * Regarding $\mathbb{E}[D_\ell]$, it is not clear on what distribution or randomness the expectation is taken. Note that $D_\ell$ is not a random variable or a function of random variables.
    * The theorem does not provide any insight for model design or understanding. There is no doubt that two nodes will be assigned to the same codeword with a higher probability if they are neighbors.
* Moreover, some other mathematical formulas are problematic as well. For example:
    * In lines 147-148, the notation $\\{\cdot\\}$ indicates that $\mathbf{Z}$ and $\mathbf{H}$ are sets, while the operations in other equations imply that they are matrices.
   * In (10), $\pi_i=-\Vert h_i-\mathbf{C}\Vert^2$ is not well defined since $h_i$ is a vector (according to (8)) but $\mathbf{C}$ is a matrix. This made the follow-up analysis meaningless.
    * Line 340 is inconsistent with (16).
* As far as I am concerned, also mentioned by Reviewer Bsb5, the technical novelty of using the Gumbel-Softmax regularization is not significant.
* The experimental results in Tables 2 and 3 indicate that the improvement of the proposed method over the strongest competitors is really tiny. There is no statistical significance analysis at all. Therefore, it is unclear why we need such a method, given that there are a few simpler alternatives.
* The motivation for raising the perplexity hasn't been justified correctly. Or in other words, there is no convincing justification for the necessity of avoiding the so-called "codebook collapse". The codebook in machine learning and signal processing is often designed to be redundant; therefore, it is not necessary to utilize all the codewords. Note that theoretically, when more codewords are used, the generalization ability becomes weaker.

Given the above issues, I have to recommend rejecting the paper.

**Reviewer Concerns:**

The major concern of Reviewer f6SV is about the presentation. This has been addressed by the rebuttal and the revision, to some extent.

The major concerns of Reviewer Bsb5 are: 1) The novelty of Gumbel–Softmax and contrastive regularization is not significant; 2) The mathematical notation is poorly presented; 3) The theoretical discussion is weak; 4) The co-assignment bound is ill-defined; 5) The ablation study is incomplete; 6) Some numerical results contradict the theoretical claim; 7) Experimental settings haven’t been clearly explained.

The reviewer mentioned (after the day of the openreview system bug) that some concerns have been addressed. However, I believe concerns 1, 4, and 5 haven't been adequately addressed.

The major concerns of Reviewer JNRk are as follows: 1) There is a gap between the theory and the method; 2) Several mathematical terms haven’t been clearly defined and explained; 3) The comparison with the baselines may not be fair; 4) The datasets used in the experiments are not large enough to show the scalability of the proposed model.

After reading the rebuttal and the revision carefully, I think that concerns 1 and 3 haven’t been addressed. In contrast, the new results further indicates that the proposed method failed to provide a significant improvement in terms of the classification accuracy compared to the baselines.

The major concerns of Reviewer 8u7x are: 1) The scalability of the proposed method hasn’t been justified; 2) Some important baselines are missing; 3) The ablation study is incomplete; 4) The “natural regularization” claim hasn’t been formally defined and discussed.

The reviewer responded to the authors’ rebuttal before the date of openreview bug and kept the positive assessment.

**Reviewer Scores:**

Reviewers f6SV and 8u7x may maintain their rating 6, while Reviewers Bsb5 and JNRk's ratings will remain negative.

---

### Decision · Program_Chairs · 2026-01-26

Reject